# Intracellular *Staphylococcus aureus* employs the cysteine protease staphopain A to induce host cell death in epithelial cells

**Kathrin Stelzner**[1], **Aziza Boyny**[1], **Tobias Hertlein**[2], **Aneta Sroka**[3], **Adriana Moldovan**[1], **Kerstin Paprotka**[1], **David Kessie**[1], **Helene Mehling**[1], **Jan Potempa**[3,4], **Knut Ohlsen**[2], **Martin J. Fraunholz**[1], **Thomas Rudel**[1]*

**1** Chair of Microbiology, University of Würzburg, Würzburg, Germany, **2** Institute for Molecular Infection Biology (IMIB), University of Würzburg, Würzburg, Germany, **3** Faculty of Biochemistry, Biophysics and Biotechnology, Jagiellonian University, Kraków, Poland, **4** Department of Oral Immunology and Infectious Diseases, University of Louisville School of Dentistry, Louisville, Kentucky, United States of America

* thomas.rudel@biozentrum.uni-wuerzburg.de

**Data Availability Statement:** All relevant data are within the manuscript and its Supporting information files.

## Abstract

*Staphylococcus aureus* is a major human pathogen, which can invade and survive in non-professional and professional phagocytes. Uptake by host cells is thought to contribute to pathogenicity and persistence of the bacterium. Upon internalization by epithelial cells, cytotoxic *S. aureus* strains can escape from the phagosome, replicate in the cytosol and induce host cell death. Here, we identified a staphylococcal cysteine protease to induce cell death after translocation of intracellular *S. aureus* into the host cell cytoplasm. We demonstrated that loss of staphopain A function leads to delayed onset of host cell death and prolonged intracellular replication of *S. aureus* in epithelial cells. Overexpression of staphopain A in a non-cytotoxic strain facilitated intracellular killing of the host cell even in the absence of detectable intracellular replication. Moreover, staphopain A contributed to efficient colonization of the lung in a mouse pneumonia model. In phagocytic cells, where intracellular *S. aureus* is exclusively localized in the phagosome, staphopain A did not contribute to cytotoxicity. Our study suggests that staphopain A is utilized by *S. aureus* to exit the epithelial host cell and thus contributes to tissue destruction and dissemination of infection.

## Author summary

*Staphylococcus aureus* is an antibiotic-resistant pathogen that emerges in hospital and community settings and can cause a variety of diseases ranging from skin abscesses to lung inflammation and blood poisoning. The bacterium can asymptomatically colonize the upper respiratory tract and skin of humans and take advantage of opportune conditions, like immunodeficiency or breached barriers, to cause infection. Although *S. aureus* was not regarded as intracellular bacterium, it can be internalized by human cells and subsequently exit the host cells by induction of cell death, which is considered to cause tissue destruction and spread of infection. The bacterial virulence factors and underlying molecular mechanisms involved in the intracellular lifestyle of *S. aureus* remain largely

**Funding:** We thank the German Research Foundation (DFG; http://www.dfg.de) for funding this project within the Transregional Research Collaborative TRR34, project C11 (K.S., M.F., T.R.) and Z3 (T.H., K.O.). This publication was funded by the German Research Foundation (DFG) and the University of Wuerzburg in the funding programme Open Access Publishing. The funders had no role in study design, data collection and analysis, decision to publish, or preparation of the manuscript.

**Competing interests:** The authors have declared that no competing interests exist.

unknown. We identified a bacterial cysteine protease to contribute to host cell death of epithelial cells mediated by intracellular *S. aureus*. Staphopain A induced killing of the host cell after translocation of the pathogen into the cell cytosol, while bacterial proliferation was not required. Further, the protease enhanced survival of the pathogen during lung infection. These findings reveal a novel, intracellular role for the bacterial protease staphopain A.

## Introduction

*Staphylococcus aureus* is a Gram-positive bacterium frequently colonizing human skin and soft tissue, primarily the anterior nares, as part of the normal microflora [1–4]. However, in hospital as well as in community settings it arises as an opportunistic pathogen causing a plethora of diseases ranging from local, superficial skin infections, wound infections and abscesses to invasive, systemic diseases like osteomyelitis, pneumonia, endocarditis or sepsis [5]. This can be largely attributed to its vast array of virulence factors [6]. In addition, the emergence and rapid spread of methicillin-resistant *S. aureus* (MRSA) strains makes this pathogen particularly difficult to treat and leads to significant morbidity and mortality worldwide [7].

Whereas *S. aureus* was originally considered an extracellular pathogen, substantial evidence exists that it is able to invade non-phagocytic mammalian cells, like epithelial and endothelial cells, osteoblasts, fibroblasts or keratinocytes [8–11], as well as to survive internalization by professional phagocytes [12–14]. Several studies demonstrate the existence of intracellular *S. aureus* in tissue and phagocytic cells *in vivo* [15–19]. Invasion of tissue cells is facilitated by numerous different bacterial adhesins and followed by escape from the bacteria-containing vacuole and cytosolic replication [20–22]. In professional phagocytes, intracellular *S. aureus* is able to resist the antimicrobial attack by the host cell and replication occurs within phagosomes. In both cell types, the pathogen eventually kills the host cell from within and a new infection cycle can be initiated. Numerous rounds of internalization and release may lead to excessive cell and tissue destruction, persistence and dissemination of infection, immune evasion and protection from antibiotic treatment [23–25].

To date, several *S. aureus* virulence factors have been linked to intracellular cytotoxicity. In non-professional phagocytes the hemolytic α-toxin was identified as a key factor mediating intracellular cytotoxicity [26,27], whereas the bi-component leukotoxin LukAB (also known as LukGH) was shown to induce cell lysis after uptake of *S. aureus* by professional phagocytes [28–31]. Some reports also connected Panton-Valentine-leukotoxin (PVL) [29,32,33] or phenol-soluble modulins (PSMs) [34,35] with host cell killing by intracellular *S. aureus*.

Beside these toxins, *S. aureus* secretes several proteases which were shown to contribute to virulence of the pathogen [36] and whose role in the intracellular lifestyle of the pathogen has not been investigated so far. *S. aureus* possesses two papain-like cysteine proteases, staphopain A (ScpA) and staphopain B (SspB), which have almost identical three-dimensional structures, despite sharing limited primary sequence identity [37,38]. Both proteases are highly conserved among *S. aureus* isolates [39] and are expressed by the operons *scp*AB and *ssp*ABC together with their endogenous inhibitors, the staphostatins ScpB and SspC, which protect the bacteria from proteolytic degradation [40–42]. Staphopain A is secreted as zymogen and activated by autolytic cleavage once outside the bacterial cell [43]. By contrast, activation of staphopain B is the result of a proteolytic cascade initiated by aureolysin-mediated cleavage and activation of the V8 protease (SspA), which in turn processes SspB [41,44]. *In vitro* experiments revealed a

very broad activity of both enzymes causing destruction of connective tissue, the evasion of host immunity and the modulation of biofilm integrity [45–51]. A staphopain A-like protease with similar functions, called EcpA, is also expressed by *S. epidermidis* and other coagulase-negative staphylococci [36,52,53]. In contrast, orthologues of the *ssp*ABC operon have only been identified in *S. warneri* [54,55].

In this study, we identify a novel role of the *S. aureus* cysteine protease staphopain A in the intracellular lifestyle of the pathogen. We demonstrate that staphopain A, but not staphopain B, plays a role in intracellular killing of epithelial host cells after phagosomal escape of *S. aureus*. Loss of function of staphopain A in a cytotoxic *S. aureus* strain led to delayed onset of host cell death in epithelial cells. Induced expression of the cysteine protease in a non-cytotoxic bacterial strain initiated an apoptosis-likemode of cell death in the infected host cells, which was dependent on the protease activity. Additionally, staphopain A favored *S. aureus* colonization of the murine lung.

## Results

### Loss of function of a cysteine protease renders intracellular *S. aureus* less cytotoxic

In order to investigate whether *S. aureus* extracellular cysteine proteases play a role in induction of host cell death by intracellular bacteria, HeLa cells infected with the community-acquired MRSA (CA-MRSA) strain JE2 were treated with E-64d, a cell-permeable irreversible inhibitor of cysteine proteases, or a solvent control. One hour post-infection all extracellular bacteria were removed by lysostaphin treatment and cell death was determined six hours post-infection by measurement of lactate dehydrogenase (LDH) release [56]. Addition of E-64d prior to infection led to a significant reduction ($p = 0.0022$) in LDH release compared to cells treated with solvent control (DMSO) (Fig 1A). This observation led to the hypothesis that either the two extracellular cysteine proteases of *S. aureus*, staphopain A (*scp*A) or staphopain B (*ssp*B), or host cell cysteine proteases might be involved in *S. aureus* cytotoxicity. To investigate the involvement of bacterial cysteine proteases single insertional gene mutants of staphopain A and B from the Nebraska transposon mutant library [57] were tested for their cytotoxic potential (Fig 1B). HeLa cells were infected with the wild type strain JE2, the transposon insertion mutants within staphopain A or staphopain B or the non-cytotoxic control strain Cowan I and six hours post-infection LDH release was measured. Whereas the wild type strain showed high levels of cytotoxicity (54.8 ± 8.8%), infection with the *scp*A transposon mutant led to a significant decrease in cell death (11.9 ± 3.4%, $p = 0.0011$). The plasma membrane damage induced by JE2 was reduced by 78% by loss of staphopain A function. The *scp*A transposon mutant still exhibited higher cytotoxicity than the control strain Cowan I (-1.3 ± 1.4%), although cytotoxicity did not differ significantly ($p = 0.1804$). By contrast, induction of cell death after infection with the staphopain B transposon mutant (44.2 ± 14.3%) was only slightly, but not significantly decreased in comparison with the wild type ($p = 0.1804$) (Fig 1B).

The different capacities of wild type and transposon mutants to induce cell death were also observed by morphological analysis of infected HeLa cells thereby corroborating our findings (Fig 1C). Whereas JE2- and JE2 *ssp*B-infected cells showed typical signs of cell death such as cell rounding, retraction of pseudopodia and detachment from the substratum six hours post-infection [58], this phenotype was not observed for cells infected with JE2 *scp*A. There, host cells remained largely adherent, reminiscent of Cowan I-infected cells. Only a closer look allowed to see that most cells infected with the *scp*A transposon mutant were filled with bacteria (Fig 1C, see arrows). These results suggest that the *S. aureus* cysteine protease staphopain A, but not staphopain B, is involved in cytotoxicity of *S. aureus*.

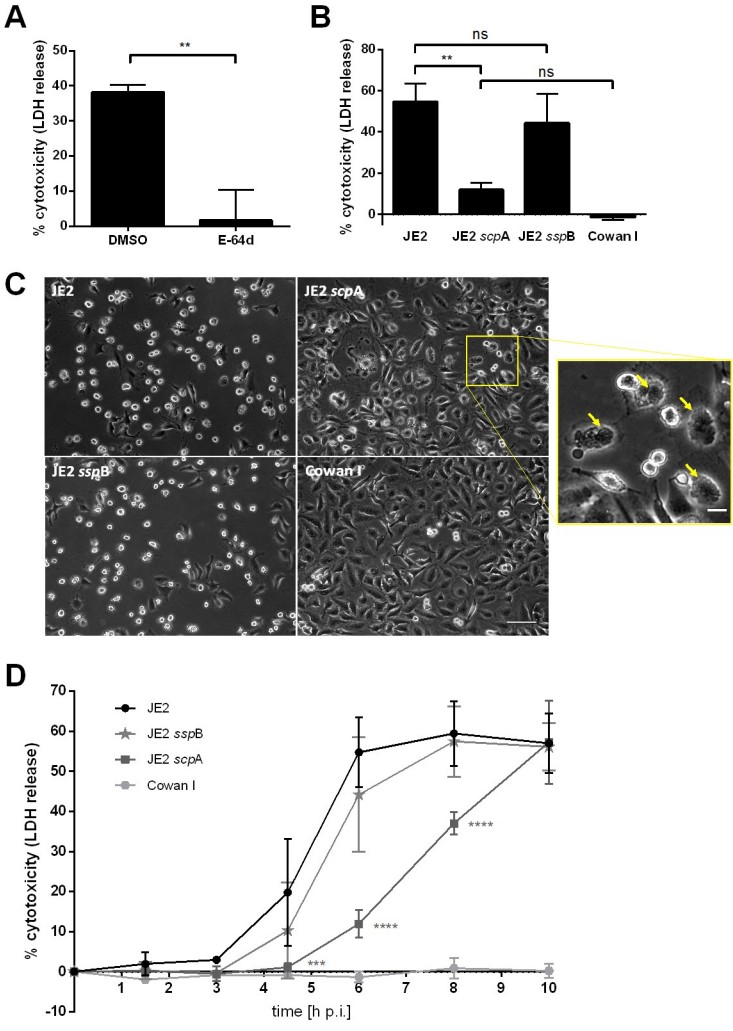

**Fig 1. *S. aureus* cysteine protease staphopain A, but not staphopain B, participates in host cell death induced by intracellular bacteria.** (A) HeLa cells were treated with 80 μM E-64d or solvent control (DMSO) 1 h prior to infection with *S. aureus* JE2. LDH release of infected cells was measured 6 h later. (B) HeLa cells were infected with *S. aureus* strains JE2 wild type, staphopain A (JE2 *scp*A) or staphopain B mutant (JE2 *ssp*B) or the non-cytotoxic strain Cowan I and cell death was assessed at 6 h p.i. by LDH release. (C) Phase contrast images of infected HeLa cells at 6 h p.i. revealed different morphologies of cells infected with the *scp*A mutant compared to wild type and *ssp*B mutant. Arrows indicate intracellular bacteria (scale bar: 100 μm, scale bar inlet: 20 μm). (D) HeLa cells were infected with wild type strain (JE2), staphopain A mutant (JE2 *scp*A), staphopain B mutant (JE2 *ssp*B) or Cowan I and cell death was assessed at 1.5, 3, 4.5, 6, 8 and 10 h p.i. by LDH assay. Statistical significance was determined by unpaired t test (A), one-way ANOVA (B) or two-way ANOVA (D) (\**P<0.01, \*\*\*P<0.001, \*\*\*\*P<0.0001).

Next, we monitored *S. aureus*-induced cytotoxicity over time. LDH release of HeLa cells infected with wild type bacteria (JE2), *scp*A mutant, *ssp*B mutant or Cowan I was measured over a period of 10 hours (Fig 1D). This assay revealed a delayed onset of cytotoxicity for the *scp*A mutant, which induced significantly less host cell death at 4.5, 6 and 8 hours post-infection when compared to wild type- or JE2 *ssp*B-infected cells. Ten hours post-infection all strains showed the same rate of cytotoxicity. Infection with the staphopain B mutant resulted in slightly, but not significantly reduced intracellular cytotoxicity.

To exclude secondary site mutations that interfere with intracellular cytotoxicity of *S. aureus* the transposon inserted in the *scp*A gene was freshly transduced into the JE2 wild type

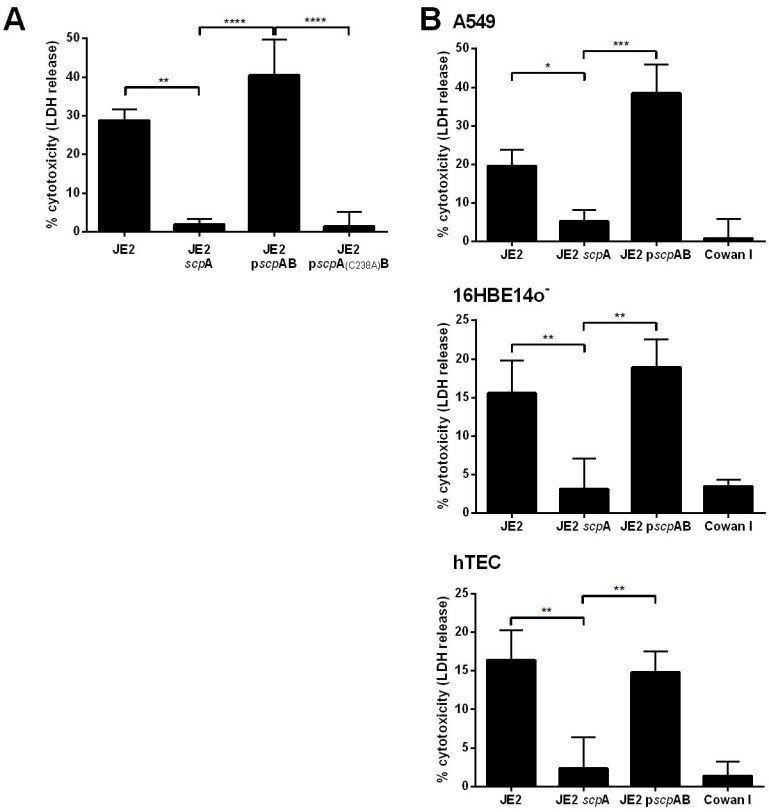

**Fig 2. Loss of staphopain A reduces intracellular cytotoxicity of *S. aureus* in different epithelial cell lines and primary cells.** (A) HeLa cells were infected with wild type strain (JE2), staphopain A mutant (JE2 *scp*A), complemented mutant (JE2 p*scp*AB) or complemented mutant with active site mutated ScpA (JE2 *scp*A$_{(C238A)}$B) and LDH release was measured 6 h p.i. (B) Further epithelial cell lines/cells such as A549, 16HBE14o⁻ and human primary tracheal epithelial cells (hTEC) were infected with JE2 wild type, staphopain A mutant (JE2 *scp*A) or complemented mutant (JE2 p*scp*AB) and cell death was assessed at 6 (A549, 16HBE14o⁻) or 8 h p.i. (hTEC) by LDH assay. Statistical significance was determined by one-way ANOVA (*$P<0.05$, **$P<0.01$, ***$P<0.001$, ****$P<0.0001$).

strain and also into the strongly cytotoxic methicillin-sensitive *S. aureus* (MSSA) strain 6850. *scp*A expression was reduced by 98% in JE2 *scp*A compared to JE2 wild type (S1A Fig). Infection of HeLa cells with the validated staphopain A mutants exhibited strongly reduced cytotoxicity (JE2 *scp*A: 1.9 ± 1.4%, 6850 *scp*A: 4.7± 5.5%) when compared to the wild type six hours post-infection (JE2: 28.8 ± 2.9%, 6850: 31.6 ± 9.0%) (Fig 2A and S1B Fig). Thereby, a loss of staphopain A led to 93% reduction of cytotoxicity in *S. aureus* JE2 and 82% reduction of cytotoxicity in *S. aureus* 6850.

In order to test whether the less cytotoxic phenotype of the staphopain A mutant can be rescued, the mutant was complemented with p*scp*AB, a plasmid expressing both staphopain A and staphostatin A under the control of their endogenous promotor. As control, the cysteine in the active site of the staphopain A protease domain was substituted by an alanine (C238A) to generate an inactive protease [43]. The *scp*A mutant containing the complementation plasmid induced significantly more LDH release than the uncomplemented mutant in JE2- or 6580-infected HeLa cells, yielding cytotoxicity levels resembling that of wild type *S. aureus* (JE2 p*scp*AB: 40.6 ± 9.2%, 6850 p*scp*AB: 40.4 ± 5.4%) (Fig 2A and S1B Fig). Cell death induced by the complemented mutant was even higher when compared to the wild type, although not significantly (JE2: $p = 0.0936$, 6850: $p = 0.4296$), which is concomitant with a higher expression of staphopain A by the high-copy plasmid (S1A Fig). By contrast, infection with

JE2 p*scp*A$_{(C238A)}$B expressing the *scp*A active site mutant resulted in a low cell death rate (1.4 ± 3.6%) reminiscent of the *scp*A mutant (Fig 2A). Further, the same kinetics of delayed cytotoxicity were observed for the active site mutant and the staphopain A mutant, while the complemented mutant behaved similarly when compared to the wild type (S1C Fig).

To further investigate the role of staphopain A in *S. aureus*-induced host cell death we performed an apoptosis detection assay by co-staining with annexin V-APC and a cell impermeable, DNA intercalating dye, 7AAD [59]. Wild type, *scp*A mutant or Cowan I-infected HeLa cells were stained with annexin V-APC and 7AAD and subsequent flow cytometric analysis confirmed that the inactivation of ScpA resulted in reduced host cell death (S1D and S1E Fig). Infection with the staphopain A mutant led to significantly fewer annexin V$^+$/7AAD$^+$ cells six hours post-infection as when compared to the wild type, whereas no difference in annexin V$^+$/7AAD$^+$ cells could be detected between infection with *scp*A mutant and Cowan I. The amount of annexin V$^+$/7AAD$^-$ and annexin V$^-$/7AAD$^+$ cells did not reveal any difference between infection with JE2, the *scp*A mutant or Cowan I. This observation indicates a late apoptotic or necrotic behavior of JE2-infected cells at six hours post-infection.

Additionally, we investigated the intracellular effect of staphopain A in other epithelial cell lines (Fig 2B). Adenocarcinomic human alveolar basal cells (A549) and immortalized human bronchial cells (16HBE14o$^-$) were infected with *S. aureus* JE2, JE2 *scp*A, JE2 p*scp*AB or Cowan I and six hours post-infection LDH release was quantified. In both cell lines cytotoxicity of the *scp*A mutant was significantly (A549: $p = 0.0371$, 16HBE14o$^-$: $p = 0.0092$) reduced by 73% or 80%, respectively, when compared to wild type. Infection with the complemented mutant could restore cytotoxicity in both cell lines. The same results were obtained in primary human tracheal epithelial cells (hTEC) (Fig 2B). Even primary cell infection with JE2 *scp*A led to significantly ($p = 0.0031$) less host cell death at early time points of infection, i.e. 8 h p.i. Mutation of staphopain A reduced cytotoxicity of intracellular *S. aureus* by 86%, whereas infection with the complemented mutant resulted in cell death rates comparable to the wild type. Phase contrast and fluorescence microscopy further illustrated these findings (S2 Fig). Epithelial cells infected with the wild type showed cell contraction at six and eight hours post-infection, respectively, whereas infection with JE2 *scp*A did not induce morphological cell alterations, although more intracellular bacteria were observed when compared to Cowan I-infected cells.

## Staphopain A inactivation leads to increased intracellular replication of *S. aureus*

In order to exclude that reduced invasion of the *scp*A mutant accounts for the observed differences in cytotoxicity, we next investigated a role for staphopain A in internalization by the host cell. HeLa cells were infected with wild type bacteria or staphopain A mutant expressing GFP. One hour post-infection, extracellular bacteria were removed by lysostaphin treatment and the percentage of infected cells was determined by flow cytometry (Fig 3A). No significant differences in the amount of host cells infected by either JE2 or JE2 *scp*A were detected ($p = 0.8823$). Similarly, CFU counts showed no significant differences in invasion of wild type and *scp*A mutant in HeLa cells (S3A Fig). Therefore, differential internalization does not account for the observed differences in cytotoxicity.

To investigate the role of staphopain A in phagosomal escape, HeLa cells expressing a fluorescent phagosomal escape reporter, YFP-CWT [13,60], were infected with JE2 wild type and *scp*A mutant and phagosomal escape was recorded by fluorescence microscopy (Fig 3B and S3C Fig and S1 Movie). The escape rate of both strains did not show significant differences ($p = 0.8747$). Similar results were obtained in 16HBE14o$^-$ cells (S3B and S3D Fig), since both

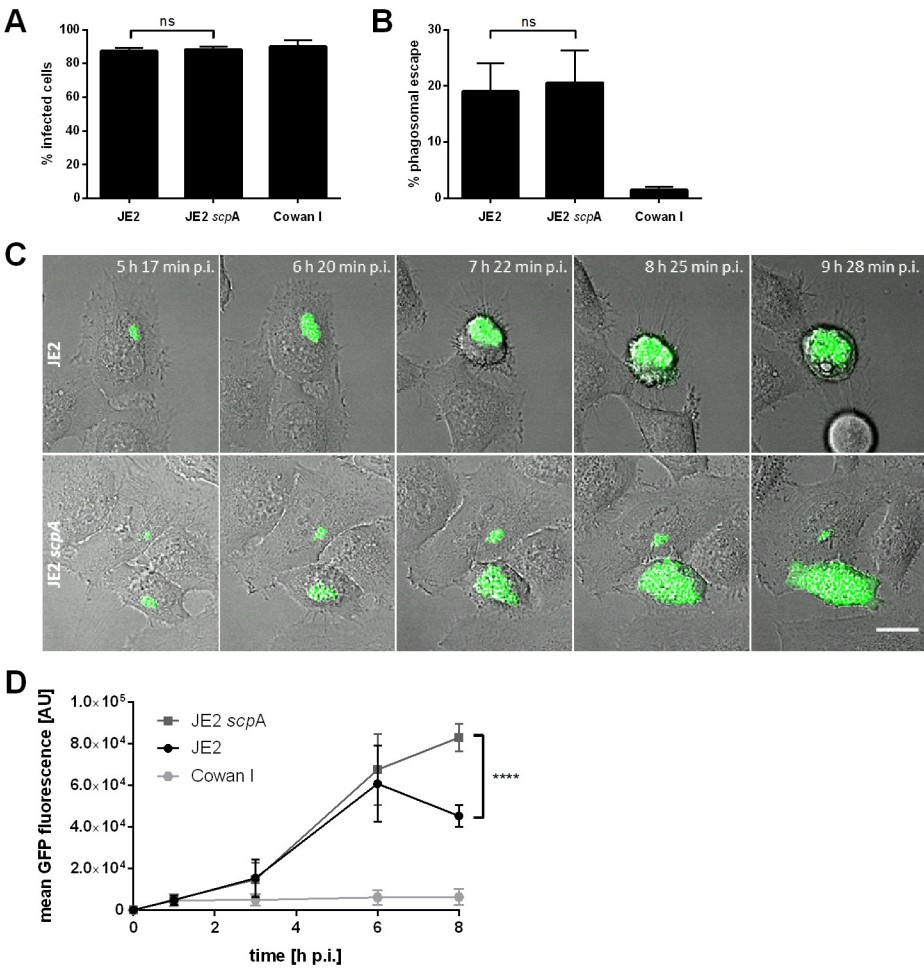

Fig 3. Staphopain A does not interfere with *S. aureus* invasion and phagosomal escape, but prevents excessive intracellular replication in the host cell cytosol. HeLa cells were infected with wild type bacteria (JE2), staphopain A mutant (JE2 *scp*A) or non-cytotoxic Cowan I expressing the fluorescent proteins GFP (A, C, D) or mRFP (B). (A) Invasion into host cells was assessed by measuring the percentage of infected, i.e. GFP-positive, cells at 1 h p.i. by flow cytometry. (B) Phagosomal escape was quantified in the marker cell line HeLa YFP-CWT 3 h p.i. by automated microscopy. (C) Live cell imaging was performed to study intracellular replication of wild type JE2 (upper panel) and staphopain A mutant (lower panel) by fluorescence microscopy (green: *S. aureus*, gray: BF, scale bar: 20 μm). (D) The mean fluorescence of infected cells was determined at 1, 3, 6 and 8 h p.i. by flow cytometry to analyze intracellular replication. Statistical significance was determined by one-way ANOVA (A, B) or two-way ANOVA (D) (****P<0.0001).

strains escaped from the phagosomes with similar efficiency. Thus, staphopain A is dispensable for *S. aureus* phagosomal escape in epithelial cells.

Escape from the phagosome is followed by intracellular replication of *S. aureus* in the host cell cytoplasm of non-phagocytic cells (S3C and S3D Fig and S1 Movie) [21,60]. Using live cell imaging we monitored replication of JE2 in HeLa cells, which was accompanied by contraction and rounding of the host cell (Fig 3C, upper panel, and S2 Movie). The *scp*A mutant displayed substantial intracellular replication, but morphological changes in the infected host cells, such as loss of adherence, were only observed later during infection (Fig 3C, lower panel, and S2 Movie). This phenotype of cells infected with the *scp*A mutant was reminiscent of phase contrast images (Fig 1C). Quantification of intracellular replication of *S. aureus* was performed by flow cytometric measurements of bacterial GFP fluorescence and revealed similar rates of JE2

and JE2 *scp*A replication up to six hours post-infection (Fig 3D). Subsequently, the *scp*A mutant continued replicating, whereas the amount of intracellular JE2 dropped as a consequence of their release from lysed cells and killing by extracellular antibiotic. Eight hours post-infection we detected a significant difference in the number of intracellular wild type and mutant bacteria (JE2: 45282 ± 5230 AFU, JE2 *scp*A: 82971 ± 6478 AFU). CFU counts of intracellular bacteria further confirmed these findings (S3E Fig). Intracellular replication correlates with the cytotoxicity of JE2 wild type and staphopain A mutant and suggests that the decrease of intracellular replication of the wild type is caused by a release of the bacteria to the medium, whereas the *scp*A mutant is less cytotoxic and therefore able to replicate longer. Indeed, we found an earlier escape of JE2 wild type bacteria from the host cells (approx. between 4.5 and 6 h p.i.) compared to the *scp*A mutant (approx. between 6 and 8 h p.i.) (S3F Fig). We therefore infected HeLa cells with *S. aureus* JE2 and JE2 *scp*A, respectively. At 1 h p.i. extracellular bacteria were removed by 30-minute lysostaphin treatment. The medium was replaced and lysostaphin was omitted for the remainder of the experiment to allow recovery of viable bacteria after exit from the host cells. Extracellular CFUs were quantified at different time points. At 8 and 10 h p.i. significantly more CFUs of JE2 wild type were recovered when compared to the *scp*A mutant (S3F Fig).

## Staphopain A expression increases during intracellular infection

We next investigated whether expression of *scp*A by intracellular *S. aureus* correlates with the intracellular cytotoxic effect of this protease. We quantified *scp*A mRNA levels during *S. aureus* intracellular infection and found that *scp*A mRNA levels were increasing over the time course of intracellular infection (6 hpi; Fig 4A), suggesting that the gene is transcribed during intracellular *S. aureus* growth. These data were corroborated by a *S. aureus* JE2 reporter strain, which contained a plasmid driving expression of superfolder GFP with the *scp*AB promoter as well as the constitutively active *sar*AP1 promoter driving mRFPmars expression (Fig 4B and 4C). Since GFP as well as mRFP expression were closely correlated, we can conclude that the *scp*AB promoter is switched on during the course of the intracellular growth of *S. aureus* (Fig 4B). Additionally, we observed increasing *scp*AB promoter activity at late stages of infection (Fig 4C). Therefore, we can conclude that *scp*A expression by intracellular *S. aureus* increases over time such as the cytotoxicity of the intracellular pathogen (Fig 1D).

## Expression of staphopain A by non-cytotoxic *S. aureus* induces host cell death

We next investigated, if expression of staphopain A in an otherwise non-cytotoxic *S. aureus* strain is sufficient to kill the host cell. Non-cytotoxic strains such as Cowan I and the laboratory strain RN4220 are not capable to escape from the phagosome (Fig 3B) [61]. Anhydrous tetracycline (AHT)-inducible expression of δ-toxin (*hld*) has previously been shown to permit phagosomal escape of *S. aureus* RN4220 in human cells, which did not cause a reduction in host cell numbers over a 24-hour period [61]. We therefore generated a transgenic strain of RN4220, which allowed for the AHT-inducible, collinear expression of δ-toxin, staphopain A, staphostatin A and the cyan-fluorescent reporter protein Cerulean. Further, we generated a variant of this plasmid, in which *scp*A was replaced with the inactive variant *scp*A$_{(C238A)}$. Both strains, RN4220 p*hld-scp*AB and RN4220 p*hld-scp*A$_{(C238A)}$B, expressed high amounts of *scp*A transcripts compared to JE2 wild type when expression was induced (S4A Fig). Additionally, the mature and active form of staphopain A was detected in the supernatant of RN4220 p*hld-scp*AB, whereas the supernatant of RN4220 p*hld-scp*A$_{(C238A)}$B contained only the inactive pro-form but not the mature form of staphopain A (S4B and S4C Fig). HeLa YFP-CWT cells were

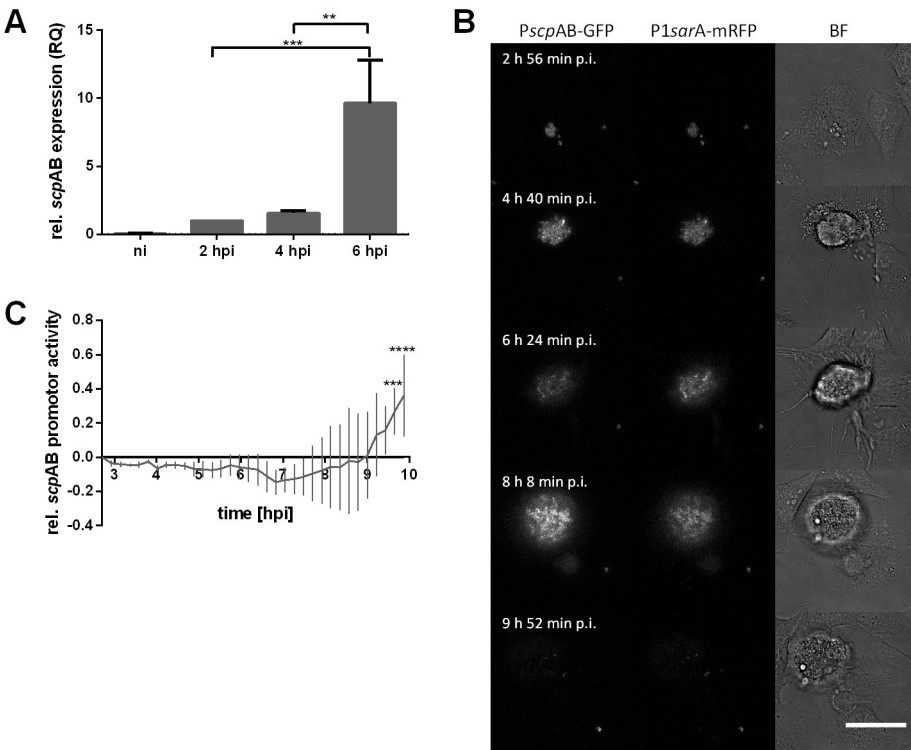

**Fig 4. Staphopain A promoter activity is increased before host cell lysis.** (A) HeLa cells were infected with *S. aureus* JE2 and RNA was isolated at 2, 4 and 6 h p.i. Transcript levels of *scp*AB were quantified by qRT-PCR and normalized to *gyr*B expression. (B, C) HeLa cells were infected with JE2 pP*scp*AB-GFP_P1*sar*A-mRFP expressing GFP under control of the *scp*AB promoter and mRFP under control of the *sar*A P1 promoter. Live cell imaging was performed to study promoter activity of the *scp*AB operon by fluorescence microscopy. (B) Stills illustrating GFP (left panel) and mRFP (middle panel) expression by intracellular *S. aureus* over the time course of infection (BF: bright field, scale bar: 40 μm). (C) mRFP fluorescence intensities. i.e. *scp*A promotor activity, were quantified over time and normalized to time point zero and *sar*A P1 promoter activity (GFP fluorescence). Statistical significance was determined by one-way ANOVA (A) or two-way ANOVA (C) (**P<0.01; ***P<0.001, ****P<0.0001).

infected with these recombinant *S. aureus* strains in the presence of AHT (Fig 5A and S3 Movie). Time-lapse microscopy demonstrated phagosomal escape of both strains, which is evidenced by accumulation of the cell-wall binding fluorescence escape reporter around the bacteria (Fig 5A, arrows). HeLa cells infected with the staphopain active site mutant RN4220 p*hld-scp*A(C238A)B remained adherent and intact, whereas cells infected with RN4220 p*hld-scp*AB often were strongly contracted (Fig 5B). Further, only infected host cells, in which *S. aureus* had escaped from the phagosome, showed this contracted phenotype (Fig 5A). Morphological changes induced by cytosolic *S. aureus* expressing functional staphopain A started with the retraction of pseudopodia, followed by strong contraction of the cell and the formation of membrane protrusions (S4D Fig). These morphological changes were similar to the ones observed in HeLa infected with *S. aureus* JE2 (compare Fig 3C and S4D Fig and S1 and S3 Movies). However, in HeLa cells infected with RN4220 p*hld-scp*AB the effects were observed within minutes after phagosomal escape, whereas cell shape in JE2-infected HeLa changed only after hours. These differences most likely result from the immediate and much higher expression of staphopain A by RN4220 p*hld-scp*AB (approx. 14,000-fold higher) (S4A Fig). Interestingly, intracellular replication of RN4220 p*hld-scp*AB was not needed to induce host cell death (S3 Movie). These data suggest that intracellular replication as well as δ-toxin expression are not responsible for the observed morphological host cell changes after escape,

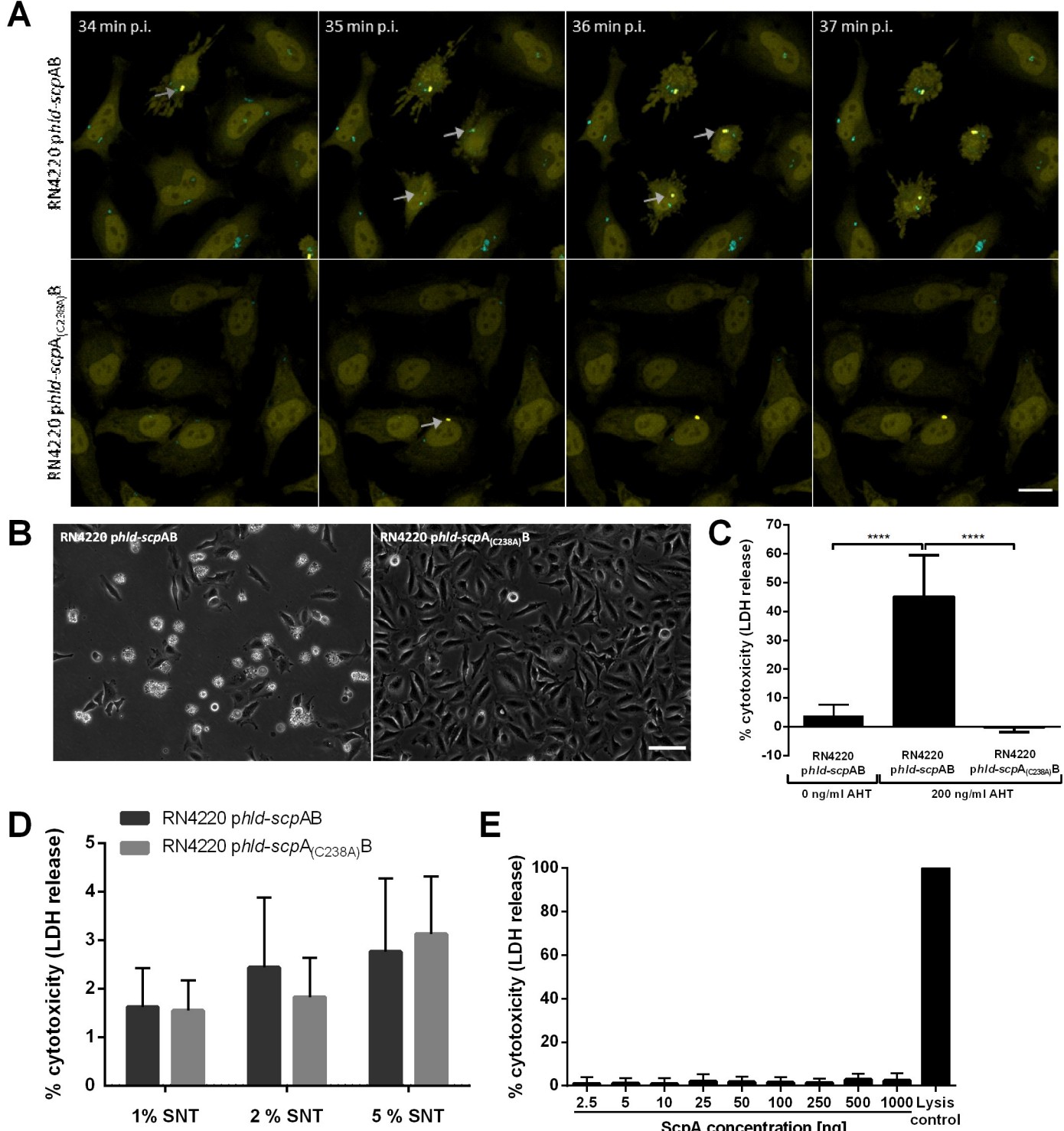

**Fig 5. Ectopic expression of staphopain A in a non-cytotoxic *S. aureus* strain induces host cell death after phagosomal escape.** (A) HeLa YFP-CWT cells were infected with *S. aureus* RN4220 p*hld-scp*AB (upper panel) or RN4220 p*hld-scp*A(C238A)B (lower panel). Live cell imaging displayed phagosomal escape (see arrows), upon which cell contraction only occurred, when a functional staphopain A was expressed (cyan: *S. aureus*, yellow: YFP-CWT, scale bar: 20 μm). (B) Phase contrast images of infected HeLa cells at 2 h p.i. to reveal different morphologies of cells infected with RN4220 p*hld-scp*AB compared to RN4220 p*hld-scp*A(C238A)B (scale bar: 100 μm). (C) LDH release of HeLa cells was quantified 6 h after infection with *S. aureus* RN4220 p*hld-scp*AB or RN4220 p*hld-scp*A(C238A)B. If indicated, 200 ng/ml AHT was added 1 h prior to infection. Overexpression of functional ScpA resulted in increased plasma membrane damage (n = 6). (D) HeLa cells were treated with 1, 2 or 5% of sterile culture supernatant (SNT) of *S. aureus* RN4220 p*hld-scp*AB or RN4220 p*hld-scp*A(C238A)B and 24 h after treatment LDH release was

determined (n = 6). (E) Treatment of HeLa cells with increasing concentrations of purified staphopain A showed no cytotoxicity. Statistical significance was determined by one-way (C, E) or two-way ANOVA (D) (****P<0.0001).

but that expression of staphopain A by *S. aureus* in the host cytoplasm leads to cell rounding and subsequent loss of adherence.

To elucidate whether the morphological changes induced by cytosolic RN4220 p*hld-scp*AB result in host cell death, an LDH assay was performed six hours post-infection (Fig 5C). Infection of HeLa cells with *S. aureus* RN4220 p*hld-scp*AB caused plasma membrane damage when toxin expression was induced by AHT (45.1 ± 14.3%). By contrast, cytotoxicity of *S. aureus* RN4220 p*hld-scp*A$_{(C238A)}$B was absent in the presence of AHT despite induction of δ-toxin (-0.2 ± 1.6%). In addition, inhibition of protease activity with E-64d in RN4220 p*hld-scp*AB infected HeLa cells with AHT treatment significantly reduced LDH release ($p$ = 0.0010) (S4E Fig). Moreover, RN4220 p*hld-scp*AB was significantly less cytotoxic towards HeLa cells when expression was not induced by AHT (4.0 ± 3.6%) compared to AHT treatment. Cytotoxicity measurements using annexin V- and 7AAD-staining further supported the role of staphopain A in host cell killing (S4F Fig).

In order to exclude a role of extracellular staphopain A in host cell death, *S. aureus* RN4220 p*hld-scp*AB and RN4220 p*hld-scp*A$_{(C238A)}$B were grown overnight in rich medium supplemented with AHT and supernatant from those cultures was sterile-filtered. 1, 2 or 5% of supernatant containing active or inactive ScpA (S4B and S4C Fig) were added onto fresh HeLa cells and an LDH assay was performed 24 hours after treatment to determine the rate of cell death (Fig 5D). A minor increase in cytotoxicity of the supernatant was observed in a concentration-dependent manner, but LDH release was comparably low in both supernatants, suggesting that extracellular staphopain A was not responsible for cell death. In addition, HeLa cells were treated with purified staphopain A using various amounts (Fig 5E), but no cytotoxicity was detected. The activity of the purified enzyme was shown (S4G Fig).

## Staphopain A induces an apoptosis-like cell death

As staphopain A-induced cell death shares morphological features with apoptosis like cell rounding, retraction of pseudopods and plasma membrane blebbing (S4D Fig) [58], we also investigated molecular characteristics of apoptosis. Annexin V- and 7AAD-staining was performed with RN4220 p*hld-scp*AB infected HeLa cells (Fig 6A and S5A Fig). Infected cells initially became annexin V-positive followed by staining with 7AAD indicating a typical staining for apoptotic cells. Apoptosis is usually also characterized by the presence of proteolytically active caspases [62]. To investigate the involvement of caspases, HeLa cells were treated with the pan-caspase inhibitor Z-VAD-fmk, infected with *S. aureus* RN4220 p*hld-scp*AB and cytotoxicity was determined six hours post-infection by measurement of LDH release (Fig 6B). Inhibition of caspases led to significantly reduced plasma membrane damage induced by infection with RN4220 p*hld-scp*AB when compared to untreated cells ($p$ = 0.0361). Moreover, time-lapse imaging revealed formation of extracellular vesicles at later time points of infection of HeLa cells with RN4220 p*hld-scp*AB (Fig 6C and S4 Movie). Activity of effector caspases was observed in those vesicles as the fluorescence of a fluorogenic substrate for activated caspase 3/7 increased. However, we found that Z-VAD-fmk also inhibited the activity of purified staphopain A in an *in vitro*-assay, although not as strongly as the cysteine protease inhibitor E-64 (S5B Fig). Further, treatment with Z-VAD-fmk did not completely abolish *S. aureus* cytotoxicity. Therefore, other cell death pathways besides apoptosis or caspase-independent apoptosis may be induced by intracellular *S. aureus*.

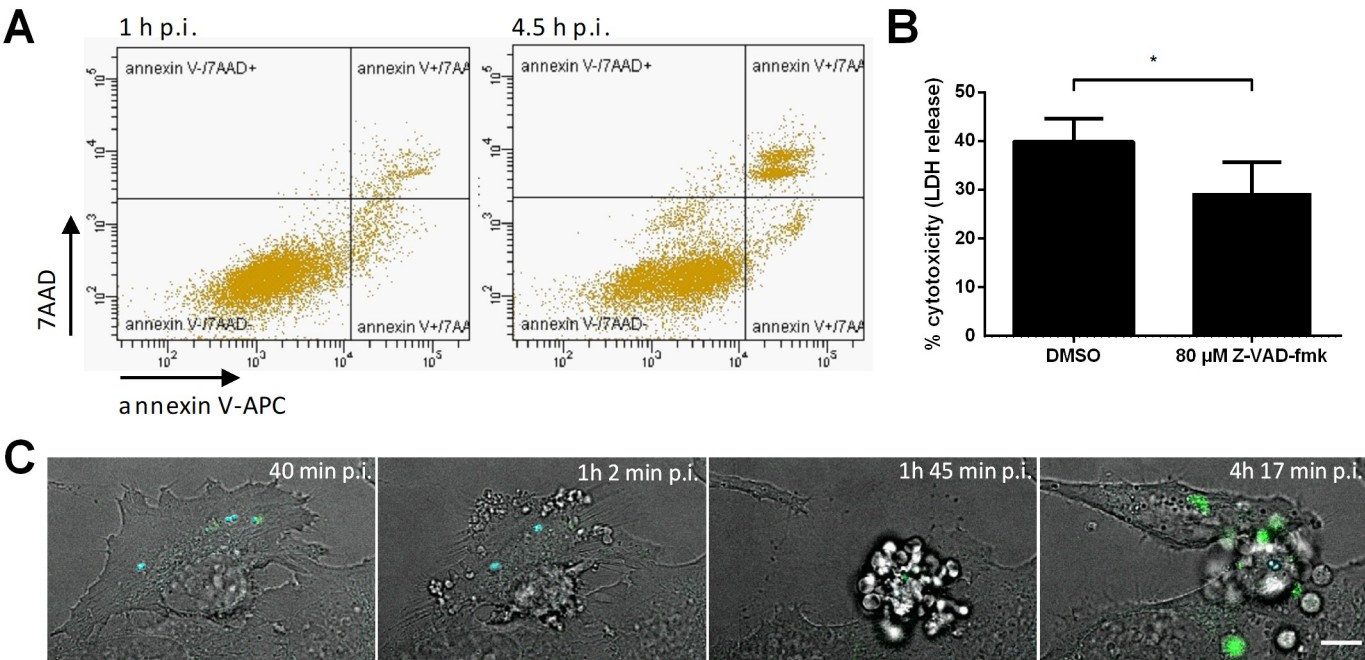

**Fig 6. Staphopain A induces an apoptosis-like cell death.** HeLa cells were infected with *S. aureus* RN4220 p*hld-scp*AB in the presence of 200 ng/ml AHT. (A) Staining with annexin V-APC and 7AAD 1 and 4.5 h p.i. and analysis by flow cytometry uncovered apoptosis characteristics of infected cells. (B) Effect of Z-VAD-fmk (80 μM) on infection-induced release of LDH 6 h p.i. was compared to solvent control (DMSO). (C) Live cell imaging of infected HeLa cells was performed to monitor the cell morphology and activation of effector caspases 3/7 (cyan: *S. aureus*, green: CellEvent Caspase3/7 Green Detection Reagent, gray: BF, scale bar: 10 μm). Statistical significance was determined by unpaired t test (*P<0.05).

## Inactivation of staphopain A leads to reduced bacterial load in murine lungs

We next tested, if staphopain A was required as a virulence factor in a murine pneumonia infection model. Mice were intranasally infected with equal doses of JE2 wild type, *scp*A mutant or a *scp*A complemented mutant strain. 48 hours after infection mice were sacrificed and bacterial CFUs from the lungs were determined by plating serial dilutions of the tissue lysate (Fig 7A). Similar amounts of JE2 and JE2 p*scp*AB were recovered from the lungs, whereas the bacterial load of the *scp*A mutant was significantly reduced (*p* = 0.0162). We excluded growth defects of JE2 *scp*A mutant (S6A Fig) as well as differential secretion of α-toxin (S6B Fig), which represents the major virulence factor in *S. aureus* induced pneumonia [63], to account for this observation. Thus, insertional inactivation of staphopain A resulted in attenuated virulence of *S. aureus* JE2 in murine lungs.

## Effects of staphopain A in immune cells

As neutrophils together with alveolar macrophages are among the first responders to combat a bacterial infection in the lung, we tested the effect of staphopain A in immune cells. *S. aureus* is known to survive phagocytosis by neutrophils and macrophages for extended periods of time [12,15,64]. Thus, we infected human primary polymorphonuclear cells (PMNs) and M-CSF-derived macrophages with *S. aureus* JE2 and JE2 *scp*A. In order to investigate intracellular effects of *S. aureus* infection, we usually apply lysostaphin to remove extracellular bacteria. However, phagocytic cells such as macrophages and neutrophils can be highly pinocytic and thus internalized antibiotics affect intracellular bacteria [14].

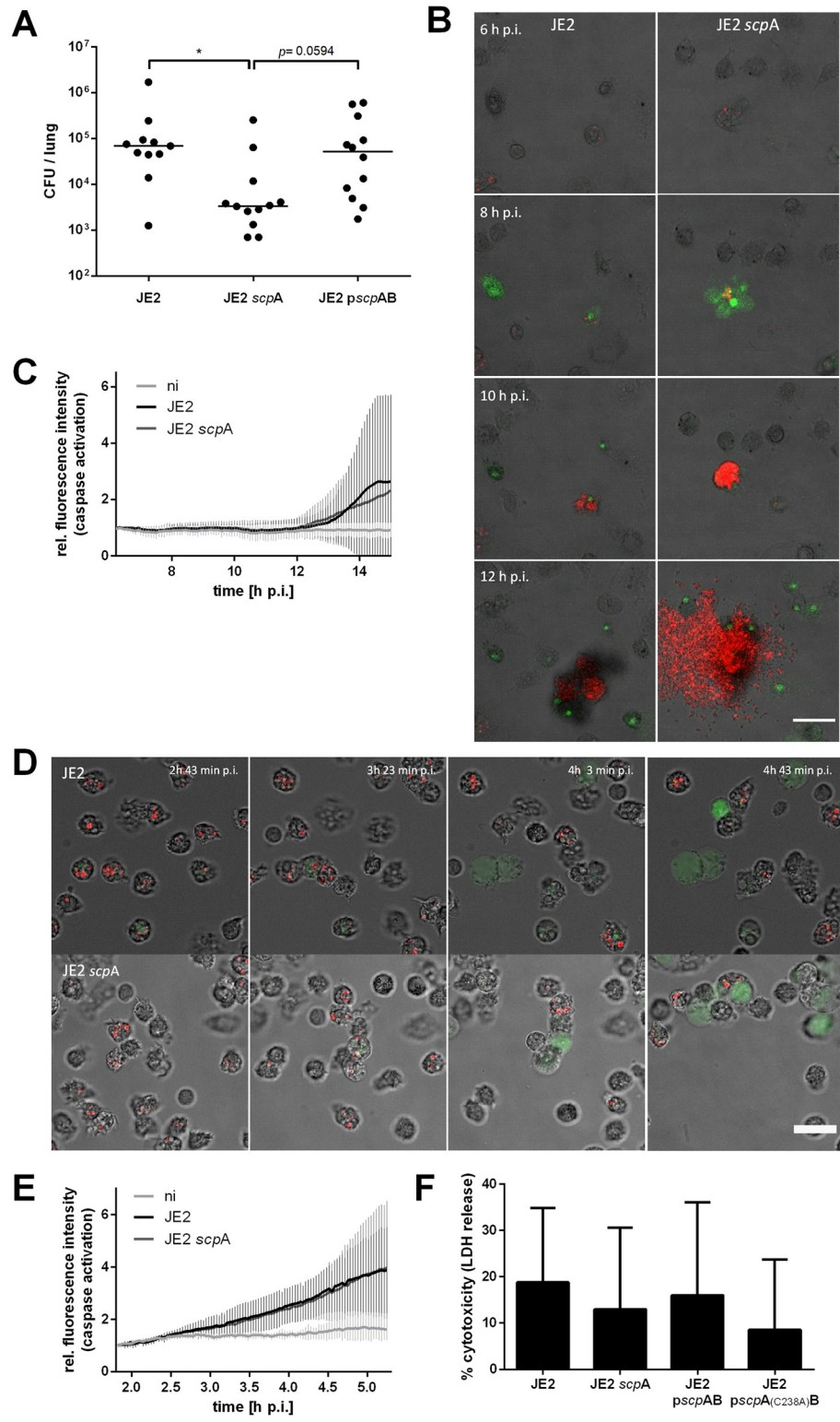

**Fig 7. Effects of staphopain A loss of function *in vivo* and in phagocytic cells.** (A) Balb/c mice were intranasally administered with wild type bacteria (JE2), *scp*A mutant (JE2 *scp*A) or complemented mutant (JE2 p*scp*AB) and bacterial CFUs were recovered from lung tissue 48 h p.i. by plating serial dilutions of the lysed tissue. The horizontal line represents the median of recovered CFUs from total lungs of mice, individual points illustrate the CFUs recovered from the lungs of one mouse. (B,C) Primary human M-CSF derived macrophages were infected with *S. aureus* JE2

mRFP and JE2 *scp*A mRFP. Live cell imaging was performed to visualize infection and quantify host cell death. (B) Stills of live cell imaging (red: *S. aureus*, green: CellEvent Caspase3/7 Green Detection Reagent, gray: BF, scale bar: 50 μm). (C) CellEvent mean fluorescence intensity was recorded for each frame, i. e. time point, and normalized to $T_0$. (D, E) Primary human PMNs were infected with *S. aureus* JE2 mRFP and JE2 *scp*A mRFP. Live cell imaging was performed to visualize infection and quantify host cell death. (D) Stills of live cell imaging (red: *S. aureus*, green: CellEvent Caspase3/7 Green Detection Reagent, gray: BF, scale bar: 20 μm). (E) CellEvent mean fluorescence intensity was recorded for each frame, i. e. time point, and normalized to $T_0$. (F) Primary human PMNs were infected with *S. aureus* JE2, JE2 *scp*A and JE2 p*scp*AB and cytotoxicity was quantified 4 h p.i. by LDH release. Statistical significance was determined by Kruskal-Wallis test (A) and one-way ANOVA (F) ($P < 0.05$).

For infection of M-CSF-derived macrophages we therefore applied a lysostaphin pulse only after infection with *S. aureus* JE2 in order to minimize uptake of the antibiotic. This experiment revealed that intracellular *S. aureus* JE2 killed only few host cells (Fig 7B and S7A Fig), although the infection rate was high. Live cell imaging using an apoptosis indicator allowed us to visualize caspase activation. Starting around 7 h p.i., very few apoptotic cells were observed, from which intracellular bacteria escaped and commenced outgrowth (S7A Fig, arrows). Eventually, all cells were killed by extracellular bacteria. Quantification of the fluorescent signal of the effector caspase activation sensor revealed no difference in cytotoxicity between *S. aureus* JE2 and JE2 *scp*A in human macrophages (Fig 7C). Induction of host cell death was only significantly detectable after 14 h of infection, when extracellular bacteria started outgrowing. Thus, the low intracellular cytotoxicity of *S. aureus* in human macrophages makes it difficult to identify the role of staphopain A in killing those cells.

Further, freshly isolated human neutrophils were infected with opsonized bacteria and treated with a low concentration of lysostaphin to remove extracellular bacteria. *S. aureus* infection activated PMNs as indicated by changes in cell morphology, such as elongation, formation of lamellipodia and membrane ruffling, as well as chemotaxis (Fig 7D and S7B Fig) [65]. Many bacteria were ingested by the activated phagocytes and we observed cell death of both *S. aureus* and PMNs. However, the phenotype of JE2 *scp*A-infected PMNs did not differ compared to wild type infection (Fig 7D). Also, cytotoxicity measured by effector caspase activation did not reveal differences between JE2- and JE2 *scp*A-infected PMNs (Fig 7E). LDH release at 4 h p.i. by infected PMNs also showed no significant effect of staphopain A on neutrophil cell death (Fig 7F), although JE2 *scp*A-infected PMNs showed a slightly lower cytotoxicity when compared to wild type and complemented mutant similar to our observations in epithelial cells (Fig 2).

## Discussion

*S. aureus* is able to invade epithelial and endothelial cells and causes cell death after phagosomal escape. Whereas phenol soluble modulins are important in translocation of endocytosed bacteria to the host cytosol [60], not much is known on the downstream processes and the respective virulence factors involved. Aside from pore-forming toxins *S. aureus* secretes a multitude of virulence factors such as lipases and proteases. It has been reported that cysteine proteases facilitate escape of intracellular pathogens from the host cell. For instance, a papain-like cysteine protease of the malarial parasite *Plasmodium falciparum* has been shown to be essential for rupture of the host cell membrane [66]. Furthermore, cysteine proteases of mammalian cells like caspases, calpains and cathepsins are associated with cell death [67]. Of the two cysteine proteases encoded by *S. aureus*, only staphopain B has been connected to *S. aureus* pathogenicity, whereas the virulence potential of staphopain A is disputed, since the mutant showed no effect in animal studies [68,69]. However, it was shown that staphopain A, for instance,

cleaves the chemokine receptor CXCR2 *in vitro* thereby blocking neutrophil activation and chemotaxis [47].

Here, we investigated the contribution of staphopain A and staphopain B to *S. aureus*-induced intracellular cytotoxicity. Treatment with the cysteine protease inhibitor E-64d prior to infection led to a significant reduction in death of host cells infected with *S. aureus* (Fig 1A), thereby suggesting an involvement of cysteine proteases of either host or pathogen origin in the pathogen-induced host cell death. We therefore tested insertional *S. aureus* mutants within either of the structural genes, *scp*A and *ssp*B, for cell death phenotypes. Interestingly, only the loss of staphopain A led to drastic reduction of cytotoxicity of both tested *S. aureus* strains, JE2 and 6850 (Fig 1B and 1D and S1B Fig). The loss of cytotoxicity in the *scp*A mutant was readily complemented in trans by reintroduction of a functional *scp*A ORF under control of its native promoter, but was absent from an active site mutant in *scp*A (C238A) [43] (Fig 2A and S1C Fig). Involvement of staphopain A in intracellular cytotoxicity of *S. aureus* was not only detected in cancer cell lines, but also in immortalized and even primary epithelial cells (Fig 2B). Our data thus demonstrate a novel role of ScpA during intracellular *S. aureus* pathogenesis, which is dependent on the proteolytic activity of staphopain A.

Bacterial extracellular proteases, such as staphopain A, process bacterial cell surface proteins, host cell receptors or extracellular matrix proteins, which are crucial for bacterial invasion [70]. In human corneal epithelial cells inhibition of staphopain A was shown to decrease adhesion and invasion of a *S. aureus* clinical isolate from a human corneal ulcer [71]. Also, increased host cell invasion is associated with enhanced *S. aureus* cytotoxicity [11]. However, staphopain A neither was involved in host cell invasion nor in phagosomal escape of *S. aureus* JE2 in epithelial cells (Fig 3A and 3B and S3A–S3D Fig and S1 Movie), another process that is implicated in intracellular *S. aureus* cytotoxicity in non-professional phagocytes [13,60,72].

Loss of staphopain A function led to a delayed onset of host cell death induced by intracellular *S. aureus* and permitted a prolonged intracellular residence and replication of the pathogen (Figs 1D and 3C and 3D). Our group previously identified a loss-of-function mutant of the pleiotropic transcriptional regulator Rsp [73]. Mutants in *rsp* demonstrate a delayed host cell death, which is reminiscent of the data obtained in the present study. Since staphopain A expression was shown to be regulated by Rsp [73], we suggest that the attenuated-cytotoxicity phenotype and the prolonged intracellular residence of the *rsp* mutant is at least partially due to diminished expression of staphopain A. Transcription of *scp*A was reported to be increased in *S. aureus* residing in THP1 macrophages [74] and we observed increased expression of staphopain A during intracellular infection in epithelial cells (Fig 4).

We next used the laboratory cloning strain *S. aureus* RN4220 and engineered it to allow controlled release of staphopain A into the host cytosol by inducible co-expression of the PSM δ-toxin, as well as *scp*AB and the fluorescent marker Cerulean. Expression of δ-toxin led to translocation of the transgenic bacteria into the host cytosol [61]. Subsequently, the strain expressing wild type *scp*A caused host cell rounding and detachment from the substratum, whereas expression of the active site mutant *scp*A (*scp*A$_{(C238A)}$) did not (Fig 5A and 5B and S4D Fig). Cell death caused by RN4220 p*hld*-*scp*AB was only detectable, when *S. aureus* escaped into the cytoplasm (Fig 5A and 5C), highlighting the dependency of *S. aureus* cytotoxicity on phagosomal escape and pointing to an intracellular target substrate of staphopain A in epithelial cells. Whether staphopain A acts alone or in concert with other bacterial factors, cannot be completely answered.

Although transgenic RN4220 escaped to the cytoplasm of infected HeLa cells, it did not grow in this environment irrespective of the presence of a functional ScpA protease (S4 Movie), illustrating that ScpA is not responsible for intracytoplasmic growth of *S. aureus*. RN4220 exhibits altered expression of virulence factors, for instance this strain shows delayed

expression of the major virulence regulator *agr* with small amounts of RNAIII and failure to translate α- and δ-toxin [75], which is based on a mutation in *agr*A [76]. Hence, the *agr* deficiency or other mutations in this cloning strain may cause the inability to replicate in the host cell cytoplasm. Besides, ectopic expression of staphopain A induces cell death rather quickly, which in turn may not allow bacterial replication (Fig 5A).

We excluded extracellular staphopain A to account for cytotoxicity to HeLa cells (Fig 5D and 5E). Interestingly, purified staphopain B induced cell death in human neutrophils and monocytes, but not in other cell types, such as macrophages and epithelial and endothelial cells [77]. We observed no cytotoxic effect of purified staphopain A on epithelial cells (Fig 5E).

The fact that overexpression of staphopain A in a non-cytotoxic *S. aureus* strain leads to strong and immediate host cell death, whereas in cytotoxic *S. aureus* strains the protease only delays *S. aureus* cytotoxicity suggests that the pathogen possesses multiple mechanisms to kill its host cell. We recently identified a staphopain A-independent cell death pathway where intracellular *S. aureus* perturbs of the host cell $Ca^{2+}$-homeostasis [78].

Inducible expression of *scp*A by *S. aureus* in HeLa cytosol indicated an apoptotic mode of cell death since the host cells first became annexin V positive and lysed only later (Fig 6A). Further, we observed morphological characteristics of apoptotic cells, such as cell rounding, retraction of pseudopods, plasma membrane blebbing and formation of extracellular vesicles reminiscent of apoptotic bodies (Fig 6C) [58,79]. However, activity of the effector caspases 3/7 during staphopain A-induced cell death was detected at rather late stages of host cell death (Fig 6C and S4 Movie). Additionally, treatment with Z-VAD-fmk could not fully abolish cytotoxicity of *S. aureus* RN4220 p*hld-scp*AB (Fig 6B) and ScpA activity was decreased by pan-caspase inhibitor treatment *in vitro* (S5B Fig). Therefore, ScpA may induce a caspase-independent apoptotic cell death.

Our results further suggest that cytotoxicity experiments using Z-VAD-fmk treating *S. aureus* infection should be carefully evaluated, because this inhibitor may affect the activity of other proteases. This finding may also explain the controversies on apoptosis induced by *S. aureus* infection [21]. Some studies describe induction of apoptosis in professional as well as non-professional phagocytes [80–83], while others observed hallmarks of both apoptosis and necrosis [14,84].

A *S. aureus* mutant lacking ten of the major extracellular proteases, including staphopain A, showed higher mortality and conversely less bacterial burden in the lungs, liver, heart and spleen, but not in brain and kidneys in a murine sepsis model and exhibited reduced bacterial loads per abscess in a murine model of skin abscess [85]. Interestingly, numbers of intracellular bacteria in professional phagocytes of whole human blood were increased for the protease-null mutant of *S. aureus* compared to the wild type [85], supporting our finding in HeLa cells where loss of staphopain A delays bacterial induced cell death and thus enables prolonged intracellular replication. However, the observed phenotype in the aforementioned study cannot be clearly assigned to staphopain A, since additional proteases were inactivated in the strain used.

The *in vivo* role of staphopain A is still debated. Inhibition of staphopain A reduced the numbers of bacteria, the corneal pathology score and the numbers of infiltrating PMNs in a *S. aureus* keratitis mouse model [71]. Inhibition of cysteine proteases by E-64 in a murine ocular infection model also resulted in reduced *S. aureus* virulence [86]. Additionally, expression of staphopain A was enhanced *in vivo* during infection in a murine osteomyelitis model compared to *in vitro* exponential growth [87]. Here, we employed a murine lung infection model to demonstrate the *in vivo* relevance of *S. aureus* staphopain A and found less bacterial burden in the lung tissue of infected mice, when staphopain A was mutated in comparison to the wild type (Fig 7A). We used the highly cytotoxic *S. aureus* USA300 derivative JE2, whereas another

study, which detected no effect of ScpA loss of function on virulence in a mouse abscess model [68], infected with low cytotoxic *S. aureus* 8325–4 [34]. Hence, the infection models of the latter study may not be suitable to reveal the contribution of staphopain A on cytotoxicity of *S. aureus in vivo*.

It was shown that *S. aureus* can survive within alveolar macrophages and neutrophils, which are the first line of defense in the lung [23]. We did not detect differences in intracellular cytotoxicity between wild type and staphopain A mutant in primary human macrophages and neutrophils (Fig 7B–7F). The intracellular lifestyle of *S. aureus* in phagocytic cells differs clearly from that in epithelial cells. Phagosomal escape is not observed in macrophages and neutrophils when compared to non-professional phagocytes [22]. As we detected cytotoxic activity of staphopain A only after phagosomal escape in HeLa cells and therefore expect localization of staphopain A target(s) in the host cell cytosol, the lack of staphopain A-mediated cytotoxic effect in phagocytic cells is reasonable.

Therefore, our finding that staphopain A contributes to increased bacterial burden in the murine lung may be explained by a faster destruction of epithelial tissue mediated by ScpA and subsequently enhanced spread of infection, whereas lack of staphopain A leads to prolonged bacterial intracellular residence in epithelial cells (Fig 3C and 3D and S2 Movie). Additionally, we cannot exclude extracellular effects of staphopain A in the murine lung. For instance, a study showed that ScpA is an important virulence factor that can impair innate immunity of the lung through degradation of lung surfactant protein A (SP-A) [88]. Cleavage of the chemokine receptor CXCR2 by staphopain A and thereby blocking of neutrophil recruitment may also play a role in *S. aureus* pathogenicity *in vivo* [47]. A better animal model to study of *S. aureus* intracellular infection is needed to distinguish extra- and intracellular effects of this pathogen *in vivo*.

In summary, we demonstrate in the present study that *S. aureus* cysteine protease staphopain A induces cell death in epithelial cells after translocation to the host cell cytoplasm. In phagocytic immune cells, where *S. aureus* only resides within phagosomes, staphopain A plays no role in cytotoxicity. We hypothesize that intracellular *S. aureus* utilizes staphopain A to induce cell death in the host cytoplasm, which in turn allows the pathogen to exit the epithelial host cell thereby leading to tissue destruction and potentially dissemination of infection. In addition, staphopain A is required for efficient colonization of the mouse lung by cytotoxic *S. aureus*, suggesting that this protease and its cytotoxic activity may be crucial for the virulence of the pathogen *in vivo*. Furthermore, we show that host cell death by intracellular *S. aureus* is not only a host cell-driven response induced by stress or cell defense, but also pathogen-driven by staphopain A.

## Methods

### Ethics statement

All animal studies were approved by the local government of Franconia, Germany (approval number 55.2 2532-2-155) and performed in strict accordance with the guidelines for animal care and experimentation of German Animal Protection Law and the DIRECTIVE 2010/63/EU of the EU. The mice were housed in individually ventilated cages under normal diet in groups of four to five throughout the experiment with ad libitum access to food and water.

Bronchial segments for hTEC isolation were obtained from three patients undergoing elective pulmonary resection. Informed consent was obtained beforehand and the study was approved by the institutional ethics committee on human research of the Julius-Maximilians-University Würzburg (vote 182/10 and 17917) and Otto-von-Guericke University Magdeburg (vote 163/17).

Experiments using human blood were approved by the Ethics Committee of the University of Würzburg (AZ 2015091401). Blood was drawn from healthy adult volunteers, who provided written informed consent.

## Bacterial culture conditions

*Escherichia coli* strains were grown in Luria-Bertani broth (LB) and *Staphylococcus aureus* strains were grown in Tryptic soy broth (TSB, Sigma), if not stated otherwise. Media were supplemented with appropriate antibiotics, when necessary, and broth cultures were grown aerobically at 37˚C overnight at 180 rpm. *E. coli* was selected on LB plates containing 100 μg/ml ampicillin and selective TSB plates for *S. aureus* were prepared using 10 μg/ml chloramphenicol and/or 5 μg/ml erythromycin (for chromosomally encoded antibiotic resistance).

## Bacterial growth curves

Bacterial growth curves were measured with a TECAN Infinite M Plex plate reader. Bacterial cultures were inoculated in triplicates to an $OD_{600nm}$ of 0.1 in 400 μl TSB and grown for 18 hours at 37˚C with orbital shaking in a 48 well microtiter plate. Absorbance at 600 nm was recorded every 10 minutes.

## Construction of bacterial strains and plasmids

All used strains, plasmids and oligonucleotides can be found in S1–S3 Tables in the supplemental material. The *S. aureus* insertional transposon mutant of staphopain A (NE1278) from the Nebraska Transposon mutant library [57] was transduced via phage φ11 into the genetic background of wild type *S. aureus* JE2 and 6850 in order to avoid secondary site mutations.

For complementation of staphopain A the *scp*AB operon including the native promotor region, 361 bp upstream of start codon, and the transcription termination signal, 253 bp downstream of the stop codon, was amplified by PCR (for primer see S3 Table) from genomic DNA of *S. aureus* JE2 and gfp$_{uvr}$ in p2085 was replaced by the generated insert using cloning sites PstI and EcoRI.

Site directed mutagenesis for $Cys_{238}$>Ala active site substitution in *scp*A was performed using the QuikChange II Site-Directed Mutagenesis Kit (Agilent Technologies) using oligonucleotides MP_*scp*A1 and MP_*scp*A2 [43].

Plasmids pmRFPmars and pGFPsf were transduced into the respective strains (see S1 Table) using phage φ11. pGFPsf was constructed by cloning sarAP1-mRFPmars from pmRFPmars into the pSK5632 backbone using cloning sites PscI and KasI and replacing mRFPmars with synthetic, codon-adapted Superfolder GFP (GFPsf, GeneArt, ThermoFisher) using cloning sites AvrII and BamHI.

For construction of p*hld-scp*AB-cerulean and p*hld-scp*A$_{(C238A)}$B-cerulean the *scp*AB operon was amplified from p*scp*AB or p*scp*A$_{(C238A)}$B, respectively, using primers *scp*AB_AvrII_fwd and *scp*AB_AvrII_rev and cloned into p*hld-hlb*-cerulean by replacing *hlb* using AvrII restriction sites [61].

For generation of a promotor-reporter strain, the plasmid pP*scp*AB-GFP_P1*sar*A-mRFP was created by amplifying the *scp*AB promotor from p*scp*AB and inserting it into p2085 using cloning sites SphI and PmeI, thereby replacing the Pxyl/tet-Promotor driving GFP expression. The *sar*A Promotor P1 and mRFP sequences, P1*sar*A-mRFP, were isolated from pmRFPmars by PstI restriction site and ligated into pP*scp*AB-GFP.

All assembled vectors were transformed into chemically competent *E. coli* DH5α and confirmed via PCR and Sanger sequencing (SeqLab, Göttingen). Subsequently, vectors were

electroporated into *S. aureus* RN4220 and, if required, further transduced into *S. aureus* JE2 or 6580 using phage φ11.

## Preparation of sterile supernatant

Bacteria were grown overnight at 180 rpm in BHI medium (Sigma Aldrich) supplemented with antibiotics and 200 ng/μl AHT, if indicated. Cultures were adjusted to an $OD_{600nm}$ of 10. Subsequently, bacterial cultures were centrifuged and the supernatant was sterile filtered (0.22 μm pore size).

## Purification of Staphopain A

*S. aureus* strain V8 (BC10 variant) was grown overnight in TSB (4 l) supplemented with β-glycerophosphate (5 mg/ml), bacterial cells were removed by centrifugation (5,000 x g, 4˚C, 20 min) and proteins in supernatant were precipitated at 4˚C by slow addition of $AmSO_4$ to 80% concentration (561 g/l) with continuous steering. After the last portion of $AmSO_4$, the sample was stirred for 60 minutes and then precipitated proteins were pelleted by centrifugation (10,000 x g, 4˚C, 30 min) and re-suspended in ice cold 50 mM sodium acetate, pH 5.5 (NaAc5.5). After extensive dialysis (24 h, against 2 l NaAc5.5 with 3 changes, 4˚C) precipitate was removed by centrifugation (20,000 x g, 4˚C, 20 min) and the supernatant passed through 0.45 μm filter. The sample was loaded on a HiPrep Q Fast Flow 16/10 column (#28-9365-43, GE Healthcare) equilibrated with NaAc5.5. The column was washed with NaAc5.5 at a flow rate of 1 ml/min and 8–10 ml fractions were collected. Staphopain A (ScpA) proteolytic activity was determined using azocasein and active fractions were pooled. Of note, staphopain B (SspB), V8 protease and aureolysin bind HiPrep Q FF and could be eluted with NaCl gradient. Next, a pool containing Staphopain A was directly loaded on a HiPrep CM Fast Flow 16/10 column (#28-9365-42, GE Healthcare) equilibrated with NaAc5.5 and the column was washed until $OD_{280nm}$ reached a baseline. Staphopain A was eluted with a NaCl gradient from 0 to 0.2 M in the total volume of 500 ml developed at the flow rate of 1 ml/min. Fractions containing proteolytic activity on azocasein in the presence of 2 mM dithiothreitol (DTT) and 5 mM EDTA were pooled and concentrated by ultrafiltration on an Amicon 10 kDa cut-of membrane and loaded on a HiLoad 16/60 Superdex 75 pg column (#28-9893-33, GE Healthcare) equilibrated with NaAc5.5 and fractions containing the proteolytic activity on azocasein were pooled and concentrated as described above. The procedure yielded up to 10 mg of >95% pure staphopain A (ScpA) (by SDS-PAGE) free of other staphylococcal proteases.

## Staphopain A activity assay

The fluorogenic substrate Z-Leu-Leu-Glu-AMC (#S-230-05M, R&D) was used to monitor staphopain A activity *in vitro*. 200 μM substrate was mixed with 50 μl 4x buffer (0.4 M sodium phosphate pH 7.4, 5 mM EDTA, freshly supplemented with 8 mM DTT) and 50 μl sterile bacterial culture supernatant in a total volume of 200 μl and transferred into a 96 well plate. The plate was incubated for 3 hours at 37˚C and mean fluorescence intensity with emission at 445 nm and excitation at 345 nm was measured with a fluorescence plate reader. For each sample values were subtracted from a blank sample (without enzyme). Activity of purified staphopain A was determined with the same protocol, except plates were incubated for 22 hours until fluorescence measurement. To investigate the inhibitory activity of E-64 (Merck) and Z-VAD-fmk (Invivogen) various concentrations of the substances were added to the substrate-buffer mix with 200 ng purified enzyme.

Alternatively, staphopain A activity was determined with azocasein (1% final concentration) as the substrate using 0.1 M Tris, 5 mM EDTA, pH 7.6 freshly supplemented with 2 mM

DTT. Briefly, enzyme sample was mixed with assay buffer in the total volume of 200 μl. After temperature was adjusted to 37˚C, 100 μl of 3% azocasein in the assay buffer was added. After one hour the reaction was stopped by addition of 200 μl of 10% ice cold trichloroacetic acid (TCA). After 10 minutes on ice precipitate was removed by centrifugation (10,000–12,000 x g, 5 min, 4˚C), 200 μl were transferred to a microplate and $OD_{360nm}$ was measured against a blank sample containing all reagents but the enzyme samples.

### Infection of epithelial cells

HeLa cells (HeLa 229, ATCC CCL-2.1) were grown in RPMI1640 medium (#72400021, ThermoFisher Scientific) and 16HBE14o⁻ [89] and A549 cells were grown in DMEM (D6429, Sigma Aldrich) supplemented with 10% FBS (Sigma Aldrich) and 1 mM sodium pyruvate (ThermoFisher Scientific) at 37˚C and 5% $CO_2$. For infection 0.8 to 1 x $10^5$ cells were seeded into 12 well microtiter plates 24 hours prior to infection. One hour prior to infection medium was renewed and, if required, treatment with 80 μM E-64d (Merck), 80 μM Z-VAD-fmk (Invivogen) or 200 ng/ml anhydrous tetracycline (AHT) was applied.

Bacterial overnight cultures were diluted to an $OD_{600nm}$ of 0.4 and incubated for one hour at 37˚C and 180 rpm to reach exponential growth phase. Then, bacteria were washed twice by centrifugation and used to infect HeLa cells at a multiplicity of infection (MOI) of 50, if not stated otherwise. After one hour co-cultivation extracellular bacteria were removed by 30 minutes treatment with 20 μg/ml lysostaphin (AMBI) followed by washing and further incubation in medium containing 2 μg/ml lysostaphin and, if required, 80 μM E-64d, 80 μM Z-VAD-fmk or 200 ng/ml AHT until the end of experiment.

### Isolation and infection of primary epithelial cells

Isolation of primary human tracheal epithelial cells (hTEC) was performed as described previously [90]. Briefly, the airway mucosa was removed mechanically from human tracheobronchial biopsies, which were subsequently placed into plastic cell culture dishes and covered with Airway Epithelial Cell Growth Medium (AECG, # PB-C-MH-350-0099, PeloBiotech). After 8–12 days hTEC grown out of the tissue pieces were collected. Infection of primary cells was performed as described above for epithelial cell lines. Except, 1.75 x $10^5$ cells were seeded into 24 well microtiter plates 24 hours prior to infection.

### Isolation, differentiation and infection of primary human macrophages

Primary human macrophages were derived from peripheral blood mononuclear cells (PBMCs) isolated from whole blood apharesis cones using the SepMate-50 system (StemCell Technologies) and Ficoll-Paque (GE Healthcare) gradient according to manufacturer's instructions.

Monocytes were purified from the PBMC fraction using the EasySep CD14+ system (StemCell Technologies) according to the manufacturer's instructions and seeded in Nunc UpCell culture plates (ThermoFisher Scientific) at a density of 5–7 x $10^6$ cells/plate in RPMI1640 (#72400054, ThermoFisher Scientific) containing 10% FBS (Sigma Aldrich) and Penicillin/ Streptomycin (ThermoFisher Scientific). Culture medium was supplemented with 50 ng/ml recombinant human macrophage colony-stimulating factor (M-CSF) (StemCell Technologies). Culture medium was replaced with fresh medium containing M-CSF and Penicillin/ Streptomycin on days 1 and 4 following monocyte isolation. Macrophages were allowed to differentiate for 7 days. On day 7, macrophages were detached from the Nunc UpCell plate by incubation at 20˚C, following manufacturer's instructions. Cells were collected by centrifugation (200 x g, 5 min), resuspended in medium without antibiotics and seeded at a

density of 1 x $10^5$ cells in 24 well microtiter plates. All infection experiments were performed on day 8.

Cells were infected at MOI 5 and plates were centrifuged for 5 minutes at 1000 rpm. After 30 minutes 20 µg/ml lysostaphin was added for 15 minutes to remove extracellular bacteria. Subsequently, cells were washed and further incubated without lysostaphin.

## Isolation and infection of primary human PMNs

Primary human polymorphonuclear leucocytes (PMNs) were isolated from heparin-anticoagulated blood from healthy adult volunteers by centrifugation over a Ficoll-Paque gradient as described previously [91]. Isolated PMNs were harvested by centrifugation at 1,000 rpm for 5 minutes and resuspended in serum-free RPMI1640. 5 x $10^5$ cells were seeded into 12 well microtiter plates, which were pre-coated with 20% normal human serum. Bacteria were prepared for infection as described above and opsonized with 10% normal human serum and tumbling for 20 minutes at 37˚C. PMNs were infected with opsonized *S. aureus* at a MOI of 10 and phagocytosis was synchronized by centrifugation at 400 x g for 10 minutes. 1 hour after infection 5 µg/ml lysostaphin was added. We were not able to remove lysostaphin, because the PMN cell population is quite heterogeneous concerning adhesion to the well plate and in our hands, centrifugation activated the infected cells.

## LDH assay

Cells were infected as described above. For epithelial cells, 1.5 hours after infection medium was replaced by RPMI1640 without phenol red containing 1% FBS and 2 µg/ml lysostaphin. PMNs were resuspended in RPMI1640 without phenol red before infection and 5 µg/ml lysostaphin was added 1 h p.i. At indicated time points after infection, medium was removed from the wells, shortly centrifuged and 100 µl of supernatant of each sample were transferred into the well of a 96 well microtiter plate in triplicates. LDH release was measured using the Cytotoxicity Detection Kit Plus (Roche) according to manufacturer's instruction. Uninfected cells served as negative control and lysed cells served as positive control.

For LDH assay with sterile supernatant, 1, 2 or 5% dilutions of sterile bacterial supernatant were prepared in RPMI1640 medium containing 1% FBS and no phenol red and added to fresh HeLa cells. 24 hours after treatment LDH release was measured as described above.

## Annexin V and 7AAD staining

HeLa cells were infected as described above. At the desired time point after infection medium, which possibly contained detached dead cells, was collected from the wells and adherent cells were detached using TrypLE (ThermoFisher Scientific). Adherent and suspension cells of each sample were pooled and after centrifugation for 5 minutes at 800 x g cells were carefully resuspended in annexin V staining buffer (RPMI1640 medium without phenol red containing 1% FBS, 2 µg/ml lysostaphin, 2 mM $CaCl_2$, 10 µl/ml annexin V-APC [BD Biosciences] and 10 µl/ml 7AAD [BD Biosciences]). After 10 minutes incubation in the dark cells were immediately analyzed by flow cytometry using a FACS Aria III (BD Biosciences) and BD FACSDiva Software (BD Biosciences). Forward and sideward scatter (FSC-A and SSC-A) were used to identify the cell population and doublet discrimination was performed via FSC-H vs. FSC-W and SSC-H vs. SSC-W gating strategy. APC or 7AAD fluorescence was measured using a 633 nm or 561 nm laser for excitation and a 660/20 nm or 610/20 nm band pass filter for detection, respectively. 10,000 events were recorded for each sample. Uninfected cells served as negative control.

### Flow cytometry-based invasion assay

HeLa cells were infected with GFP expressing *S. aureus* strains and prepared for flow cytometry as described above. Detached cells were resuspended in fresh medium without phenol red containing 1% FBS and 2 μg/ml lysostaphin one hour post infection, after 10 minutes treatment with 20 μg/ml lysostaphin to remove extracellular bacteria. For determining invasion, the percentage of GFP-positive cells representing the infected cells was measured by flow cytometry using a FACS Aria III (BD Biosciences). Gating and analysis were performed as described above. GFP fluorescence was measured using a 488 nm laser and a 530/30 nm band pass filter for detection. Uninfected cells were used to determine autofluorescence of the cells and signals above this value were defined as infected.

### Flow cytometry-based intracellular replication assay

Cells were infected with GFP expressing strains and prepared for flow cytometry as described above. Intracellular replication was determined by measurement of GFP fluorescence (arbitrary units, AU) of the infected cells 1, 3, 6 and 8 hours after infection by flow cytometry using a FACS Aria III (BD Biosciences). The intensity of GFP fluorescence corresponds to the amount of intracellular bacteria. Gating and analysis were performed as described above. GFP fluorescence was measured using a 488nm laser and a 530/30 nm band pass filter for detection. Uninfected cells served as negative control.

Bacterial invasion into the host cells as well as intracellular replication were additionally analyzed by counting bacterial colony forming units (CFU) recovered from infected cells.

### Phagosomal escape assay

Phagosomal escape was determined as described previously with minor modifications [13]. Briefly, HeLa YFP-CWT cells were infected with mRFP-expressing bacterial strains at a MOI of 10 in a 24 well μ-plate (ibidi). After synchronization of infection by centrifugation, infected cells were incubated for 1 hour to allow bacterial invasion. Subsequently, a 30 minute-treatment with 20 μg/ml lysostaphin removed extracellular bacteria, after which the cells were washed and medium with 2 μg/ml lysostaphin was added. Three hours after infection, cells were washed, fixed with 4% paraformaldehyde overnight at 4°C, permeabilized with 0.1% Triton X-100 and nuclei were stained with Hoechst 34580. Images were acquired with an Operetta automated microscopy system (Perkin-Elmer) and analyzed with the included Harmony Software. Co-localization of YFP-CWT and mRFP signals indicated phagosomal escape.

### CFU assay

Infection was performed as described above. After one hour bacteria-host cell co-cultivation, extracellular bacteria were removed by treatment with 20 μg/ml lysostaphin.

Bacterial invasion was assessed by employing a 10-minute lysostaphin treatment, followed by washing with sterile PBS and subsequent osmotic shock-mediated lysis in 1 ml alkaline water (pH 11) for 5 minutes at room temperature. Dilution series of the lysate were plated on tryptic soy agar (TSA) and incubated overnight at 37°C to enumerate bacterial numbers. Colony forming units (CFU) are given per ml representing total bacterial numbers per well. Invasion rates were calculated as bacterial numbers recovered after the lysostaphin pulse (70 min post-infection) relative to the initial bacterial inoculum.

Samples subjected to intracellular replication assessment were incubated with 20 μg/ml lysostaphin for 30 minutes, followed by washing and further incubation in medium containing

2 μg/ml lysostaphin until the end of the experiment. Intracellular replication was determined by plating dilution series as described above, at 3, 6 and 8 hours after infection.

## Live cell imaging

Human cells were seeded in 8 well chamber μ-slides (ibidi) 24 hours prior to infection, except for PMNs, which were seeded shortly before infection. Infection with fluorescent protein-expressing bacterial strains was performed at a MOI of 5 for epithelial cells and macrophages and at MOI10 for PMNs as described above. Time-lapse imaging of the samples was performed in imaging medium (RPMI1640 without phenol red containing 10% FBS and, if required, lysostaphin) on a Leica TCS SP5 confocal microscope using a 40x (Leica HC PL APO, NA = 1.3) or 63x (Leica HCX PL APO, NA = 1.3–0.6) oil immersion objective. The μ-slides were transferred to a pre-warmed live cell incubation chamber (Life Imaging Systems) surrounding the confocal microscope and perfusion with 5% $CO_2$ in a humidified atmosphere and a temperature of 37˚C was applied during imaging. LAS AF software (Leica) was used for setting adjustment and image acquisition. All images were acquired at a resolution of 1024x1024 pixels and recorded in 8-bit mode at predefined time intervals. Z-stacks were imaged with a step size of 0.4 μm. All image-processing steps were performed using Fiji [92]. For detection of effector caspase activity 20 μM CellEvent Caspase 3/7 Green Detection Reagent (ThermoFisher Scientific) was added to the infected cells prior to imaging. For quantification of fluorescence intensities raw imaging data was used. Fluorescence intensities (mean of RFU) were measured and data were normalized to time point zero ($R_0$) obtaining relative fluorescence values. For single cell analysis one region of interest covering one cell (ROI) was defined for all recorded time frames.

Phase contrast microscopic images of live, infected cells were acquired with a LEICA DMR microscope connected to a SPOT camera using a 10x objective (Leica HC PL FLUOTAR, NA = 0.32) and VisiView software (Visitron).

## Murine pneumonia infection

Overnight cultures of *S. aureus* strains in BHI medium were diluted to a final $OD_{600nm}$ of 0.05 in 50 ml fresh BHI medium and grown for 3.5 hours at 37˚C. All growth media for *S. aureus* strains containing plasmids were supplemented with appropriate antibiotics. After centrifugation, the bacterial pellet was resuspended in BHI with 20% glycerol, aliquoted and stored at -80˚C. For infection, aliquots were thawed, washed twice with PBS and adjusted to the desired infection inoculum of 2 x $10e^8$ CFU/20 μL. A sample was plated on TSB agar plates to confirm the correct bacterial concentration. Female Balb/c mice (6 weeks, Janvier Labs, Le Genest-Saint-Isle, France) were intranasally instilled with the infection dose. Mice were scored twice a day and sacrificed after 48 hours of infection. Lungs were harvested, homogenized and plated in serial dilutions on TSB agar plates in order to measure the bacterial burden in the individual organs.

## RNA isolation

*S. aureus* cell pellets from overnight cultures were shock-frozen in liquid nitrogen and RNA was isolated according to Lasa et al. [93]. For isolation of RNA from infected samples, cells were covered in RNAprotect Cell Reagent (Qiagen) and RNA was isolated using the RNeasy Mini Kit (Qiagen). Therefore, the cell pellet was resuspended in RLT buffer supplemented with β-mercaptoethanol. The samples were transferred to Lysing Matrix B tubes (MP Biomedicals) and lysed for 45 sec at 6 m/sec in a FastPrep FP120 cell disruptor. After centrifugation the supernatant was transferred to a fresh tube and RNA isolation was continued as recommended

in the manufacturer's protocol. Residual DNA in RNA samples was digested by using the TURBO DNA-*free* DNase kit (Invitrogen). DNA digestion was verified by PCR using qRT-primers against *gyr*B.

## Quantitative real-time PCR

For synthesis of cDNA the RevertAid First Stand cDNA kit (Thermo Scientific) was used. 1000 ng RNA were mixed with random hexamer primers. qRT-PCRs were performed on a StepOne Plus PCR system (Applied Biosystems) in a 96 well plate format. Reaction mixes consisted of GreenMasterMix (Genaxxon), 300 nM primer and 100 ng cDNA. Results were analyzed using the $2^{-\Delta\Delta Ct}$ method [94]. Relative gene expression was normalized to expression of the housekeeping gene of gyrase subunit B (*gyr*B).

## SDS-PAGE and Immunoblotting

For SDS-PAGE bacterial secreted proteins from sterile supernatant were precipitated overnight at -20˚C in 25% trichloroacetic acid (TCA). After centrifugation at 4˚C for 30 minutes the pellet was washed twice with ice-cold acetone. The dried pellet was resuspended in 2x Laemmli buffer (100 mM Tris/HCl (pH 6.8), 20% glycerol, 4% SDS, 1.5% β-mercaptoethanol, 0.004% bromophenol blue) and immediately incubated at 95˚C for 10 minutes for protein denaturation. Proteins were separated via gel electrophoresis on 12% polyacrylamide gel and transferred to a PVDF membrane (Sigma Aldrich) using a semi-dry blotting system. The PVDF membrane was incubated for 1 hour in blocking solution (5% human serum in 1x TBS-T) and overnight at 4˚C with the first antibody (diluted in blocking solution). The staphopain A antibody (ABIN967004, antibodies online) was diluted 1:500 in blocking solution. α-toxin was detected with the respective antibody (S7531, Sigma Aldrich) diluted 1:2000 in blocking solution. Primary antibodies were detected with a horseradish peroxidase (HRP)-conjugated secondary antibody (170–6515, Biorad, 1:3000 in 1x TBS-T with 5% non-fat dry milk) using enhanced chemiluminescence (ECL) and an Intas imaging system (Intas Science Imaging).

## Statistical analyses

Data were analyzed using GraphPad Prism Software (GraphPad Software, Version 6.01). For statistical analysis three biological replicates were performed, if not indicated otherwise. All data are presented as means with standard deviation (SD). P-values ≤0.05 were considered significant. Pairwise comparisons were assessed using unpaired Student's t-test. Analysis of variance (ANOVA) was performed to determine whether a group of means was significantly different from each other. ANOVA was performed with Tukey's post-hoc analysis for defining individual differences and Dunn's multiple comparison test was applied for Kruskal-Wallis test.

## Supporting information

**S1 Fig. Staphopain A contributes to bacterial induced host cell death in *S. aureus* JE2 and 6850.** (A) RNA was isolated from bacterial overnight cultures and expression of *scp*A in JE2 *scp*A, JE2 p*hld-scp*AB and JE2 p*hld-scp*A(C238A)B was determined by qRT-PCR and normalized to *scp*A expression of JE2 wild type. (B) HeLa cells were infected with *S. aureus* 6850 wild type, transposon mutant of staphopain A (6850 *scp*A) or complemented mutant (6850 p*scp*AB) and cytotoxicity was determined by LDH release 6 h p.i. (C) HeLa cells were infected with wild type strain (JE2), staphopain A mutant (JE2 *scp*A), complemented mutant (JE2 p*scp*AB),

complemented mutant with active site mutation (JE2 *scp*A(C238A)B) or Cowan I and cell death was assessed at 1.5, 3, 6, 12 and 24 h p.i. by LDH assay. (D, E) HeLa cells were infected with wild type (JE2), staphopain A mutant (JE2 *scp*A) or Cowan I and apoptotic cells were determined by annexin V-APC and 7AAD staining and flow cytometric analysis 6 h p.i. Statistical significance was determined by one-way ANOVA (A, B) or two-way ANOVA (D) ($^*$P<0.05, $^{**}$P<0.01, $^{****}$P<0.0001).

(TIF)

**S2 Fig. Staphopain A cytotoxicity in epithelial cells.** Microscopic images of A549, 16HBE14o⁻ and hTEC cells infected with JE2 wild type, JE2 *scp*A or Cowan I expressing mRFP at 6 h p.i. (A549, 16HBE14o⁻) or 8 h p.i. (hTEC) (gray: PC, red: *S. aureus*, scale bar: 20 μm).

(TIF)

**S3 Fig. Invasion, phagosomal escape and intracellular replication of *S. aureus* JE2 and JE2 *scp*A in HeLa and 16HBE14o⁻ cells.** (A) HeLa cells were infected with JE2, JE2 *scp*A and Cowan I and invasion into HeLa cells was determined by quantifying intracellular CFUs at 1 h p.i. (B) Phagosomal escape was quantified in the marker cell line 16HBE14o⁻ YFP-CWT 3 h p.i. by automated microscopy after infection with mRFP-expressing bacteria. (C) Infected HeLa YFP-CWT cells were imaged over time to visualize phagosomal escape of JE2 wild type (upper panel) and staphopain A mutant (JE2 *scp*A, lower panel) (red: *S. aureus*, yellow: YFP-CWT, gray: BF, scale bar: 20 μm). (D) Live cell imaging of 16HBE14o⁻ YFP-CWT cells infected with *S. aureus* JE2 mRFP or JE2 *scp*A mRFP (scale: 20 μm). (E) HeLa cells were infected with JE2, JE2 *scp*A and Cowan I and intracellular bacterial CFUs were quantified at 1, 3, 6 and 8 h p.i. (F) HeLa cells were infected with JE2, JE2 *scp*A and Cowan I and only a lysostaphin-pulse was applied to remove extracellular bacteria after invasion. Bacteria, which escaped from the host cell, were quantified by CFU plating at 3, 4.5, 6, 8 and 10 h p.i. Statistical significance was determined by unpaired t-test (B), one-way ANOVA (A) or two-way ANOVA comparing JE2- to JE2 *scp*A-infected samples (E, F) ($^*$P<0.05, $^{**}$P<0.01, $^{****}$P<0.0001).

(TIF)

**S4 Fig. Intracellular expression of staphopain A but not extracellular addition of the protease leads to cell death.** (A) RNA was isolated from bacterial overnight cultures and expression of *scp*A in RN4200 p*hld-scp*AB and RN4220 p*hld-scp*A(C238A)B was determined by qRT-PCR and normalized to *scp*A expression of JE2 wild type. (B) *S. aureus* RN4220 p*hld-scp*AB or RN4220 p*hld-scp*A(C238A)B were grown overnight with or without addition of 200 ng/ml AHT. Proteins of the sterile culture supernatant were precipitated and western blot was performed to detect staphopain A. Expression of functional ScpA was detected as the mature protein (triplet ranging from 17 to 20 kDa), while expression of a non-functional staphopain A led to accumulation of proScpA (ca. 40 kDa) [43]. (C) Proteolytic activity of staphopain A was measured from sterile culture supernatant of *S. aureus* RN4220 p*hld-scp*AB or RN4220 p*hld-scp*A(C238A)B. (D) Imaging of HeLa cells infected with *S. aureus* RN4220 p*hld-scp*AB visualized cell contraction over time (cyan: *S. aureus*, gray: BF, scale bar: 20 μm). (E) The effect of E-64d (80 μM) treatment on cytotoxicity of RN4220 p*hld-scp*AB infected HeLa cells was determined by quantification of LDH release 6 h p.i. and compared to solvent control (DMSO). (F) HeLa cells were infected with *S. aureus* RN4220 p*hld-scp*AB or RN4220 p*hld-scp*A(C238A)B with or without addition of 200 ng/ml AHT prior to infection. 4.5 h p.i. cells were stained with annexin V-APC and 7AAD and analyzed by flow cytometry. (G) Proteolytic activity of staphopain A was measured from increasing concentrations of the purified enzyme. Statistical significance was determined by unpaired t test (C, E) or two-way ANOVA (F) ($^*$P<0.05, $^{**}$P<0.01,

***P<0.001, ****P<0.0001).
(TIF)

**S5 Fig. Cytotoxicity of *S. aureus* RN4220 p*hld-scp*AB and effect of protease inhibitors on staphopain A proteolytic activity.** (A) HeLa cells were infected with *S. aureus* RN4220 p*hld-scp*AB in the presence of 200 ng/ml AHT and 1 and 4.5 h p.i. cells were stained with annexin V-APC and 7AAD and analyzed by flow cytometry. (B) The inhibitors E-64 and Z-VAD-fmk reduce staphopain A proteolytic activity in a concentration-dependent manner. DMSO was used as solvent control. Statistical significance was determined by two-way ANOVA (****P<0.0001).
(TIF)

**S6 Fig. Growth curves of *S. aureus* JE2, JE2 *scp*A and JE2 p*scp*AB and α-toxin expression of *S. aureus* JE2 and JE2 *scp*A.** (A) *S. aureus* JE2, JE2 *scp*A and JE2 p*scp*AB were grown in TSB in 48-well plates for 18 h and the optical density at 600 nm was measured every 10 minutes. (B) *S. aureus* JE2 and JE2 *scp*A were grown overnight, proteins of the sterile culture supernatant were precipitated and western blot was performed to detect α-toxin (Hla).
(TIF)

**S7 Fig. Effects of staphopain A in phagocytic cells.** (A) Primary human M-CSF derived macrophages (A) or PMNs (B) were infected with *S. aureus* JE2 mRFP and JE2 *scp*A mRFP. Live cell imaging was performed to visualize infection (red: *S. aureus*, green: CellEvent Caspase3/7 Green Detection Reagent, gray: BF, scale bar: 200 μm (A)/50 μm (B)).
(TIF)

**S1 Movie. Phagosomal escape of *S. aureus* JE2 and JE2 *scp*A.** HeLa YFP-CWT cells were infected with JE2 wild type (left panel) or staphopain a mutant (JE2 *scp*A, right panel) and imaged over time to visualize phagosomal escape (red: *S. aureus*, yellow: YFP-CWT, gray: BF).
(AVI)

**S2 Movie. Intracellular replication of *S. aureus* JE2 and JE2 *scp*A.** HeLa cells were infected with *S. aureus* JE2 GFP (left panel) or JE2 *scp*A GFP (right panel) and time-lapse imaging was performed (green: *S. aureus*, gray: BF).
(AVI)

**S3 Movie. Phagosomal escape of *S. aureus* RN4220 p*hld-scp*AB or RN4220 p*hld-scp*A$_{(C238A)}$B.** HeLa YFP-CWT cells were infected with *S. aureus* p*hld-scp*AB (left panel) or RN4220 p*hld-scp*A$_{(C238A)}$B (right panel) and time-lapse imaging was performed (cyan: *S. aureus*, yellow: YFP-CWT).
(AVI)

**S4 Movie. Activation of effector caspases in HeLa cells infected with *S. aureus* RN4220 p*hld-scp*AB.** HeLa cells were infected with *S. aureus* RN4220 p*hld-scp*AB and a fluorogenic caspase3/7-substrate was added. Live cell imaging was performed to monitor the effect of staphopain A expression on cell morphology and activation of effectors caspases (cyan: *S. aureus*, green: CellEvent Caspase3/7 Green Detection Reagent, gray: BF).
(MP4)

**S1 Table. Bacterial strains used in this study.**
(PDF)

**S2 Table. Plasmids used in this study.**
(PDF)

**S3 Table. Oligonucleotides used in this study.**
(PDF)

## Acknowledgments

We are grateful to Sudip Das for cloning of constructs, and Ursula Eilers and the Core Unit Functional Genomics (University Würzburg) for support with Operetta Imaging. Maria Steinke (University Hospital Würzburg and Fraunhofer IGB—Unit Würzburg) is thanked for technical advice during the establishment of the hTEC infection model. We thank Jan-Peter Hildebrandt (University Greifswald) for providing 16HBE14o⁻ cells. Rosemarie Bott (University Würzburg) is gratefully acknowledged for taking blood samples and we thank all volunteers for blood donation.

## Author Contributions

**Conceptualization:** Kathrin Stelzner, Tobias Hertlein, Aneta Sroka, Jan Potempa, Knut Ohlsen, Martin J. Fraunholz, Thomas Rudel.

**Formal analysis:** Kathrin Stelzner, Aziza Boyny, Tobias Hertlein, Adriana Moldovan, Kerstin Paprotka.

**Investigation:** Kathrin Stelzner, Aziza Boyny, Tobias Hertlein, Aneta Sroka, Adriana Moldovan, Kerstin Paprotka, David Kessie, Helene Mehling.

**Writing – original draft:** Kathrin Stelzner, Martin J. Fraunholz, Thomas Rudel.

**Writing – review & editing:** Kathrin Stelzner, Martin J. Fraunholz, Thomas Rudel.

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
