## [Decision Letter · Decision Letter 0]

19 Jul 2021

Dear Dr. Rudel,

Thank you very much for submitting your manuscript "Intracellular Staphylococcusaureus employs the cysteine protease staphopain A to induce host cell death in epithelial cells" for consideration at PLOS Pathogens. As with all papers reviewed by the journal, your manuscript was reviewed by members of the editorial board and by several independent reviewers. The reviewers appreciated the attention to an important topic. Based on the reviews, we are likely to accept this manuscript for publication, providing that you modify the manuscript according to the review recommendations.

While two reviewers who reviewed the manuscript for the second time suggested acceptance, a third reviewer, who was invited because a previous reviewer was not available anymore, had some major concerns regarding the mode of cell death and the fact that the in-vivo experiments do not allow conclusions about intra- vs. extracellular toxicity. While I understand that in-vivo mechanistic evidence of that sort is difficult to obtain, I would like you to experimentally address in a revision at least the issue that reviewer 3 had about the mode of cell death (pyroptosis) as well as answer to that reviewer's additional points.

Sincerely,

Michael Otto

Section Editor

PLOS Pathogens

Michael Otto

Section Editor

PLOS Pathogens

Kasturi Haldar

Editor-in-Chief

PLOS Pathogens

orcid.org/0000-0001-5065-158X

Michael Malim

Editor-in-Chief

PLOS Pathogens

orcid.org/0000-0002-7699-2064

Reviewer Comments (if any, and for reference):

Reviewer's Responses to Questions

**Part I - Summary**

Reviewer #1: This manuscript is a resubmission of previous work that describe a novel intracellular role of staphopain A in S. aureus pathogenesis.

Although the authors did not explore additional read outs in the mouse model of pneumonia as requested, the authors performed several new experiments that assessed the role of staphopain A in S. aureus pathogenesis in other cell types (this was greatly appreciated). An especially interesting result was that primary human macrophages and neutrophils were resistant to the effects of staphopain A cytotoxicity compared to epithelial cells. While the molecular mechanism for this specificity is unknown, identifying the intracellular target(s) as well as understanding how this relates to enhanced pathogenesis in vivo will be of particular interest.

Reviewer #2: In this study, the authors interrogate the role of cysteine proteases in the intracellular lifestyle of S. aureus. They report that staphopain A plays a role in intracellular killing of the host cell after S. aureus escapes the phagosome. Inactivation of staphopain A in a cytotoxic S. aureus strain delayed onset of host cell death in epithelial cells whereas Inducible expression of the cysteine protease in a non-cytotoxic bacterial strain initiated apoptotic cell death. Finally, the authors report that staphylopain is important in a S. aureus lung colonization model. The manuscript is clear and well put together. The experiments are incisive, and the conclusions are supported by the data. I have a few minor suggestions or questions that can be addressed with text modifications.

Reviewer #3: This paper describes the role of the cystein protease scpA in the cellular toxicity of S. aureus once it has been released in the cytosol from non professional phagocytes. The loss of this protein prolongs the survival of the bacteria in the cytosol of the cells, where they continue to multiply.

The experimental design is well constructed and the results are convincing, with appropriate controls included in each experiment.

The discoveries are original and significantly contribute to the understanding of the factors governing the ability of S. aureus to survive inside the cells or to escape from them.

The main weakness is that the authors did not manage to identify the type of cell death associated with the activity of this enzyme neither its substrate(s) in the cytosol. These two elements are probably closely related.

**Part II – Major Issues: Key Experiments Required for Acceptance**

Reviewer #1: NA

Reviewer #2: None

Reviewer #3: 1. A key observation that is not exploited is the fact that cell toxicity is delayed in the absence of the cystein protease scpA but not reduced (figure 1). This observation should be better integrated in the interpretation of the mechanisms.

2. Another cell death mechanism compatible with Annexin V labeling and permeation of an intercalating agent is pyroptosis, which also involves caspase activation. This mechanism has not been explored here although it has been described for S. aureus.

3. In vivo data do not allow to distinguish between toxicity mediated by extracellular or intracellular bacteria and are thus difficult to reconcile with the role of scpA in intracellular survival.

**Part III – Minor Issues: Editorial and Data Presentation Modifications**

Reviewer #1: NA

Reviewer #2: It is not clear why the authors used the murine pneumonia model to test the importance of staphopain in vivo, there are murine models that more accurately represent S. aureus replication. Perhaps some text could be added to the manuscript to justify this decision?

Is Rsp responsible for the altered expression of staphopain observed in Figure 4?

Reviewer #3: 1. The way the data on immune cells are explained in the text is rather confusing, the discussion is much clearer in this respect.

2. How were normalized bacterial counts ? The scale is in CFU/ml (fig S3 E, for example), but per ml of what ? the number of cells is decreasing during the experiment.

3. Any idea about the factor(s) that could activate the expression of scpA in the cytosol ? Is this a question of pH ?

4. Figure 5. What is the concentration of scpA that can be found intracellularly ? Is there any evidence that it is in the range of concentrations of purified protein investigated here ?

PLOS authors have the option to publish the peer review history of their article (what does this mean?). If published, this will include your full peer review and any attached files.

Reviewer #1: No

Reviewer #2: No

Reviewer #3: No

Figure Files:

Data Requirements:

Reproducibility:

References:

---

## [Decision Letter · Decision Letter 1]

7 Aug 2021

Dear Dr. Rudel,

We are pleased to inform you that your manuscript 'Intracellular Staphylococcusaureus employs the cysteine protease staphopain A to induce host cell death in epithelial cells' has been provisionally accepted for publication in PLOS Pathogens.

Best regards,

Michael Otto

Section Editor

PLOS Pathogens

Michael Otto

Section Editor

PLOS Pathogens

Kasturi Haldar

Editor-in-Chief

PLOS Pathogens

orcid.org/0000-0001-5065-158X

Michael Malim

Editor-in-Chief

PLOS Pathogens

orcid.org/0000-0002-7699-2064

Reviewer Comments (if any, and for reference):

Reviewer's Responses to Questions

**Part I - Summary**

Reviewer #3: My comments have been satisfactorily addressed by the authors. I have no additional comments.

**Part II – Major Issues: Key Experiments Required for Acceptance**

Reviewer #3: none

**Part III – Minor Issues: Editorial and Data Presentation Modifications**

Reviewer #3: none

PLOS authors have the option to publish the peer review history of their article (what does this mean?). If published, this will include your full peer review and any attached files.

Reviewer #3: No

---

## [Editor Report · Acceptance letter]

23 Aug 2021

Dear Dr. Rudel,

We are delighted to inform you that your manuscript, "Intracellular Staphylococcusaureus employs the cysteine protease staphopain A to induce host cell death in epithelial cells," has been formally accepted for publication in PLOS Pathogens.

Best regards,

Kasturi Haldar

Editor-in-Chief

PLOS Pathogens

orcid.org/0000-0001-5065-158X

Michael Malim

Editor-in-Chief

PLOS Pathogens

orcid.org/0000-0002-7699-2064